# Adaptive partitioning of a gene locus to the nuclear envelope in *Saccharomyces cerevisiae* is driven by polymer-polymer phase separation

Lidice González[1], Daniel Kolbin[2], Christian Trahan [3], Célia Jeronimo[3], François Robert [3,4,5], Marlene Oeffinger[1,3,4], Kerry Bloom[2] & Stephen W. Michnick [1] ✉

Partitioning of active gene loci to the nuclear envelope (NE) is a mechanism by which organisms increase the speed of adaptation and metabolic robustness to fluctuating resources in the environment. In the yeast *Saccharomyces cerevisiae*, adaptation to nutrient depletion or other stresses, manifests as relocalization of active gene loci from nucleoplasm to the NE, resulting in more efficient transport and translation of mRNA. The mechanism by which this partitioning occurs remains a mystery. Here, we demonstrate that the yeast inositol depletion-responsive gene locus *INO1* partitions to the nuclear envelope, driven by local histone acetylation-induced polymer-polymer phase separation from the nucleoplasmic phase. This demixing is consistent with recent evidence for chromatin phase separation by acetylation-mediated dissolution of multivalent histone association and fits a physical model where increased bending stiffness of acetylated chromatin polymer causes its phase separation from de-acetylated chromatin. Increased chromatin spring stiffness could explain nucleation of transcriptional machinery at active gene loci.

Genes relocalize to specific regions inside the nucleus depending on whether they are active or repressed, including the nuclear periphery or the periphery of their chromosomal territories, a phenomenon observed in many different eukaryotes from budding yeast to mammalian cells[1–13]. This has been particularly well studied for *Saccharomyces cerevisiae* gene loci, for which several stress response genes, including nutrient related genes such as the *GAL* loci and heat shock response genes, translocate to the nuclear periphery upon activation[1,3,8,9]. These genes remain at the periphery after returning to their repressed condition, a form of transcriptional memory, which

primes the genes to be more rapidly transcribed, translocated to the cytosol and translated, as nutrients vary between being depleted and enriched in a fluctuating environment[11,14].

The *INO1* gene has been among the most studied to determine a molecular mechanism for gene locus translocation[1,12,15]. Efforts to understand locus localization to the nuclear envelope (NE) have focused on attempts to identify protein-protein or protein-DNA interactions that could account for physical tethering of chromatin to the NE[11,12]. Thus, interactions of component proteins of the Nuclear Pore Complex (NPC) or associated components such as subunits of the

[1]Département de Biochimie, Université de Montréal, C.P. 6128, Succursale centre-ville, Montréal, QC H3C 3J7, Canada. [2]Department of Biology, University of North Carolina at Chapel Hill, Chapel Hill, NC 27599, USA. [3]Institut de recherches cliniques de Montréal, 110 Avenue des Pins Ouest, Montréal, QC H2W 1R7, Canada. [4]Faculty of Medicine, Division of Experimental Medicine, McGill University, Montréal, QC H3A 1A3, Canada. [5]Département de Médecine, Faculté de Médecine, Université de Montréal, 2900 Boul. Édouard-Montpetit, Montréal QC, H3T 1J4, Canada. ✉e-mail: stephen.michnick@umontreal.ca

SAGA complex have been suggested to induce, or their deletion has been shown to prevent *INO1* translocation[11,12]. However, so far, there are no direct evidence that these proteins are involved in physical tethering of the *INO1* locus to the NE. Furthermore, 5′ *cis* elements in the *INO1* locus called Gene Recruitment Sequences (GRS) or "Zip Codes", have been shown to be essential to *INO1* and now other loci translocations[11–13]. It has been proposed that GRSs could be anchoring sites for the tethering of gene loci to the NE via protein-DNA or protein-protein interactions[11–13]. To date, however, evidence of interactions between GRSs and proteins at the NE have not been reported. Furthermore, studies of the statistical distributions of translocated genes among populations of cells, as well as gene tracking measurements, show that instead of being bound at the NE, as are, for instance, the NPC or Spindle Pole Body, activated gene loci are simply statistically more tightly distributed towards the NE[8,9,12,16]. These observations led us to postulate that active gene locus translocation could be the result of differential partitioning of active gene loci to the periphery by a local phase transition of the chromatin caused by histone acetylation and other modifications, resulting in increased rates of histone exchange and associated decompaction of the chromatin. Here we present evidence that the GRSs are sites of association of chromatin remodeling complexes, including HATs, which catalyze histone H3 acetylation. Consequently, there is an increase in H3 exchange between nucleosomes and nucleoplasm within and surrounding the activated *INO1* locus. The resulting decompaction of chromatin causes phase separation of the *INO1* locus from the compacted chromatin found in the nucleoplasm to the decompacted chromatin adjacent to the nuclear envelope.

## Results

Translocation of the activated *INO1* locus and decompaction of its chromatin were studied as follows: first, we obtained a yeast strain in which 128 LacO binding motifs are integrated into the gene and expression of the LacO binding protein LacI is fused to GFP[12]. Furthermore, reference markers for the nuclear envelope (Nup49-GFP) and nucleolus (Nop1-mCherry) are also expressed. This strain could then be used to monitor and compare the statistical distribution of *INO1* in cells grown in the presence (inactive *INO1*) or absence (active *INO1*) of inositol (Fig. 1a, b). Next, we could monitor the phase transition of active chromatin based on the observation that the consequent decompaction of the chromatin, is reflected in changes in its mechanical properties and geometric organization[17,18]. These changes can be understood as follows: first, the persistence length ($L_p$) of DNA, as for all polymers, is the length over which a thermal perturbation would cause it to bend. The $L_p$ for naked DNA and transcriptionally active chromatin is smaller than that of chromatin on small lenght scales (5–50 nm versus 170–220 nm)[19–23] given that these forms of DNA have more degrees of freedom than compact, inactive chromatin[17,18]. Notably, the opposite is true over long length scales. This increase in degrees of freedom implies a higher number of configurations that can be explored per unit time by naked DNA or active chromatin, so its movements are more confined compared to chromatin. Consequently, chromatin behaves as an "entropic spring" of increasing spring stiffness as it transitions from compact to decompacted states[17,18]. The spring constant of a polymer such as chromatin ($k_s$) is inversely proportional to $L_p$ (Eq. (1)) and consequently $k_s$ is predicted to increase as chromatin decompacts. Furthermore, its trajectories through space are shorter, a characteristic that is determined from the radius of confinement ($R_c$) of the chromatin[17,18]. Such a model is highly simplified, but for our purposes, measuring these values for the *INO1* gene locus would serve to quantify material properties of the chromatin and how these change under different conditions. Values for $k_s$ and $R_c$ can be determined from measurements of mean-square displacement (MSD) and localization coordinates of the GFP-labeled LacO array embedded in the

*INO1* gene locus[17].

$$k_s = \frac{k_B T}{L_p^2} \tag{1}$$

## Deletion of GRS I and II prevents localization of active *INO1* to the nuclear envelope

The relocalization of the active *INO1* locus to the nuclear periphery is prevented in strains in which the GRSI, and II of the *INO1* locus are both deleted (hereafter referred to as the GRSI, IIΔ double mutant)[24]. To access the effects of GRS deletions on the statistical distribution of the *INO1* locus in large populations of cells, we used a high-resolution computational approach that generates probabilistic maps of the locus distribution in the nucleus (Fig. 1a, b)[16]. Using the LacI-GFP-labeled LacO array as reporter for the *INO1* locus and, Nop1-mCherry and Nup49-GFP as nucleolus and nuclear envelope reference markers, respectively, we mapped the distribution of the *INO1* locus with respect to these landmarks for the *wild type* and single and double mutants of the GRS sites under repressing (+ inositol) and activating (− inositol) conditions.

Consistent with previous studies, the repressed *INO1* locus was distributed throughout the nucleoplasm, but the activated *INO1* locus was mainly confined to a specific region in the vicinity of the NE (Fig. 1c, d)[12]. Single GRS deletion mutants reduced and the double mutant GRSI, IIΔ prevented partitioning of the active *INO1* locus towards the NE, as reported previously (Fig. 1e–g)[24]. The *INO1* locus was distributed towards, but not fixed at the NE[12]. The effects of the mutants do not reflect changes in transcription of *INO1* as disrupting transcription was previously shown to have no effect on *INO1* relocalization[14].

The activated *INO1* locus shows a characteristic spatial distribution at the NE that perhaps reflects its position in the middle of the 3′ arm of Chromosome X. NE localization of other activated gene loci also reflects their chromosome positions, including the *GAL1-10* locus, which is located closer to the centromere region in Chromosome II and thus close to the Spindle Pole Body[16] and the *HXK1* gene locus, located in a subtelomeric region of Chromosome VI, that is localized towards the NE whether active or repressed, but with a more compact distribution when activated[9].

## Deletion of GRSs prevents decompaction of the active *INO1* locus

We measured differences in the mechanical spring stiffness of the *INO1* locus in the repressed and activated states. To do this, we tracked the GFP-LacI-labeled LacO array in the *INO1* locus for 1 min at intervals of 500 milliseconds and calculated MSD values in each case. The slope and plateaus of the MSD trajectories for the *INO1* locus were markedly reduced under activating versus repressed conditions (Fig. 2a). In addition, loci closer to the NE appear to be excluded from the interior of the nucleoplasm as observed before for the *GAL1-10* genes[8], which also contain GRS sequences required for relocalization of the *GAL1-10* loci to the NE[13].

We then used established relationships between MSD statistics and $R_c$ and $k_s$ to calculate their values as follows:[17] $R_c$ could be calculated directly from the MSD plateaus (See Methods, Eq. (3))[17,25]. $R_c$ could independently be determined from the standard deviation of spot positions, σ, and the average squared deviation from the mean position, $\langle \Delta r_0^2 \rangle$, by applying the equipartition theorem (See Methods, Eq. (4))[17,26,27].

The *INO1* locus in *wild type* strains (Fig. 2b) and single GRS mutant cells (Fig. 2c, d) showed significantly larger $R_c$ for the repressed versus the activated state. The GRSI, IIΔ double deletion, however, resulted in no significant differences in $R_c$ between repressed and activated states (Fig. 2e). These results were also consistent with statistical distributions of loci in populations of cells, demonstrating that the observed changes in $R_c$ in single cells are pervasive in the population.

Using the equipartition theorem and standard deviation (σ) for the distribution of steps from the mean we calculated $k_s$ (See Methods, Eq. (5))[17,26]. The $k_s$ of the *INO1* loci significantly increased in cells with active versus repressed loci in both *wild type* (Fig. 2f) and the GRS single deletion strains (Fig. 2g, h), suggesting a stiffening of decompacted chromatin. There was no significant change in $k_s$ in the GRSI, IIΔ double deletion strain under activating conditions (Fig. 2i).

### The active *INO1* locus behaves as an elastic filament

As discussed above, the increase in spring stiffness of decompacted chromatin reflects in a decrease in the rate and amplitude of *INO1* locus expansion-contraction[17]. This behavior can be visualized with the LacI-GFP-labeled LacO array and quantified by fitting an ellipsoid function to the GFP signal imaged at super-resolution by Structured Illumination Microscopy and calculating the aspect ratio between long and short axes of the ellipsoid (Fig. 2j). Consistent with predicted behavior, the *INO1* locus changes shape with higher frequencies and amplitudes in the repressed versus the activated states (Fig. 2k). These observations are also consistent with previous results in which a decrease in $L_p$

caused by H3 histone depletion resulted in a decrease in the rate and amplitude of expansion of the chromatin[17].

### Activation of the *INO1* locus increases rates of exchange of histone H3

To test whether the changes in *INO1* locus mechanical properties follow from an increase in histone exchange, we used a strain to probe changes in histone H3 nucleosome exchange in *wild type* and GRSI, IIΔ double mutant strains. In this strain histones H3 and H4 genes are deleted from the genome and replaced with both constitutively and inducible expressed copies (Fig. 3a)[28]. In a galactose inducible cassette, H3 is fused to a FLAG-tag, such that when expressed can be used to quantify newly synthesized H3 that is incorporated into nucleosomes at specific positions by Chromatin Immunoprecipitation followed by real time PCR (ChIP-qPCR) to measure nucleosome exchange in different chromatin regions under different conditions. To avoid the contribution of histone incorporation due to DNA replication, cells are also arrested in G1 by the addition of α-factor. We probed H3-FLAG incorporation in several regions in the *INO1* promoter and Open Reading Frame (ORF), plus sites

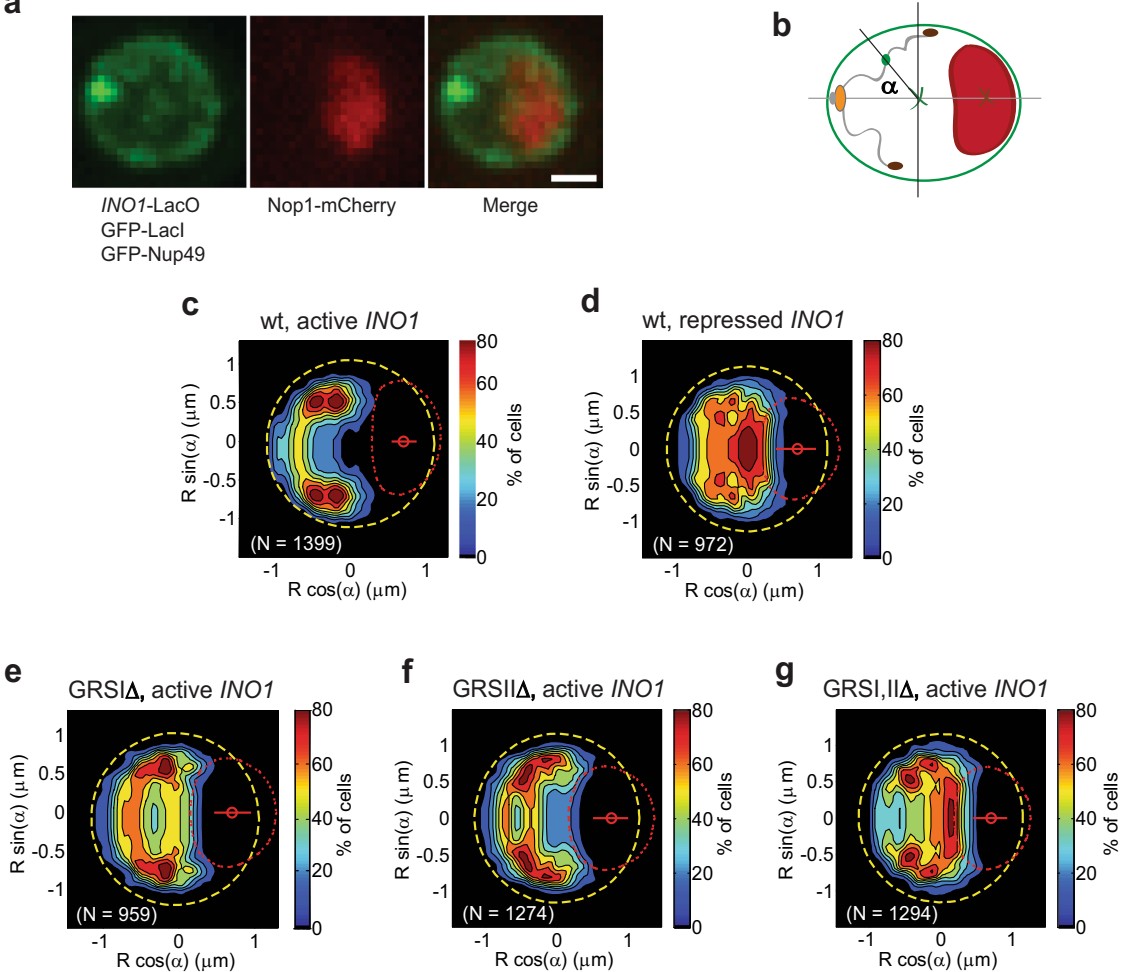

**Fig. 1 | Activated *INO1* NE localization is disrupted by the deletion of both GRS I and II. a** Representative photomicrographs of yeast nuclei containing the LacO repressor array integrated into the *INO1* locus and expressing GFP-LacI allows for visualization of dynamics in the locus with GFP-Nup49-labeled NE and Nop1-mCherry-labeled nucleolus as spatial reference markers (maximum projection of 250 nm Z stacks). Scale bar represents 1 μm. These results are representative of three independent experiments. **b** Yeast nucleus landmarks used as coordinates for this analysis. NE (green ellipsoid), nucleolus (red ellipsoid) and nucleus center (green x). The gray lines represent the central axes of the ellipsoid; α, angle from the

longer central axis. Orange, red, and green blobs connected by a trace represent the spindle pole body, telomeres and gene locus, respectively. **c** *INO1* probability maps obtained from the analysis of 1399 nuclei of *wild type* cells grown in the absence of inositol (active), and **d** 972 nuclei of *wild type* cells in the presence of inositol 100 μM (repressed). Dashed yellow line, NE; dashed red line, position of the nucleolus; the color scale represents the probability density distribution of the locus inside the nucleus. **e**–**g** *INO1* probability maps obtained from the analysis of 959 (GRSIΔ), 1274 (GRSIIΔ) and 1294 (GRSI, IIΔ) nuclei in cells in which GRS I, GRS II or both GRS I and II are deleted, respectively, and that were grown in the absence of inositol (active).

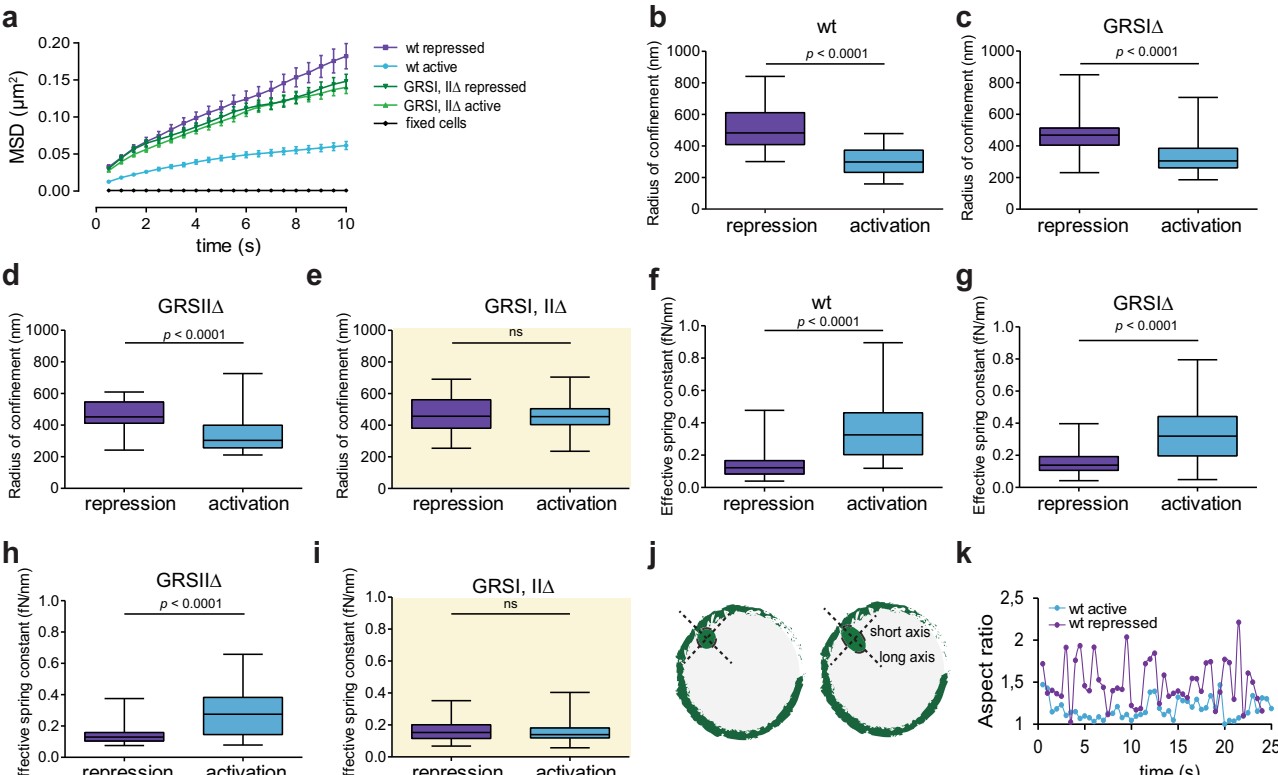

**Fig. 2 | *INO1* dynamics and material properties depend on the presence of both GRS I and II. a** MSD curves for *INO1* locus in *wild type* (purple and blue lines, *n* = 38 and 34 cells, respectively) and the GRSI, II∆ (dark and light green lines, *n* = 51 and 45 cells, respectively) strains under activating and repressed conditions, and for a population of fixed cells (black line, *n* = 8 cells). Data are presented as mean values +/− SEM. **b**−**d** Radius of confinement ($R_c$) significantly decreases in *wild type*, GRSI∆, and GRSII∆ cells from repressed (purple) to activated (blue) conditions. **e** $R_c$ for GRSI, II∆ mutant cells under repressing (purple) and activating (blue) conditions show no significant differences. **f**−**h** Spring constant ($k_s$) for *wild type*, GRSI∆, and GRSII∆ cells increases significantly from repressing (purple) to activating (blue) conditions. **i** $k_s$ for GRSI, II∆ mutant shows no significant difference between

repressing (purple) and activating (blue) conditions. Source data are provided as a Source Data file. **j** Schematic of *INO1* locus size measurements of aspect ratio variation over time. Green circles represent nuclei of two cells showing loci spots with different long and short axis ratios. **k** Aspect ratio change over time for two *INO1* loci in cells grown under repressed (purple) or activated (blue) conditions measured by Structured Illumination Microscopy. In the box and whisker plots (**b**−**i**), the median is indicated as middle line, 25th and 75th percentile as boxes and the whiskers represent minimum and maximum values; *n* = 38 and 34 cells, respectively (**b**, **f**), *n* = 46 and 51 cells, respectively (**c**, **g**), *n* = 31 cells (**d**, **h**), *n* = 51 and 45 cells, respectively (**e**, **i**). Data were analyzed by unpaired two-sided Student's *t* test in **b**−**i**.

in upstream (*SNA3* and *DAS1*) and downstream (*VPS35*) genes to determine whether changes in histone exchange are confined to the *INO1* locus (Fig. 3b, Supplementary Fig. 1).

ChIP-qPCR results with an anti-FLAG antibody showed higher incorporation of newly synthesized histone H3 at the *INO1* promoter in cells under activated versus repressed conditions for the *INO1* promoter and ORF regions (Fig. 3c). Results for repressed cells were consistent with genome-wide ChIP-chip data[28], validating our method. For the GRSI, II∆ double mutant, significant reductions in H3 exchange under activated conditions were observed in the *INO1* locus promoter at the GRS I site and in the *SNA3* promoter and coding sequence where GRS II is located but not in the *INO1* ORF sequences (Fig. 3c). These results suggest that GRS I and II may be HAT and/or other chromatin remodeling complex association sites mediating H3 exchange in topologically adjacent nucleosomes. The fact that H3 exchange is not affected in the *INO1* ORF sequence is consistent with the fact that there are other known remodeling enzymes that bind near this region. Thus, consistent with ours and previous results, an increase in nucleosome histone H3 exchange at the *INO1* locus chromatin is associated with an increase in its spring stiffness reflecting decompaction of the chromatin[17].

### Depletion of yeast HATs prevents active *INO1* locus partitioning to the nuclear envelope

The increase in H3 exchange at the *INO1* locus is consistent with histone post-translational modifications, and specifically lysine

acetylation by histone acetyltransferases (HATs)[18,28–31]. We therefore hypothesized that the GRSs could be association sites for HATs and other chromatin remodeling complexes that alter the rates of histone-nucleosome exchange. Evidence for specific functions of HATs and histone deacetylases in controlling active *INO1* partitioning towards the NE exists. For instance, the Rpd3(L) histone deacetylase blocks partitioning of the *INO1* locus towards the NE and deletion of the gene encoding this deacetylase results in increased acetylation of nucleosome-associated histone complexes in the promoter of *INO1*[15]. Additionally, deleting subunits of the SAGA HAT complex decreases active *INO1* locus partitioning to the NE[11], also observed for the *GAL* loci[8].

Since both GRS I & II, plus other unidentified GRS sites could be associated with HATs, we attempted to identify HATs associated with the *INO1* locus by screening for loss of activated *INO1* locus NE localization in cells in which an auxin-inducible degron (AID) coding sequence was integrated 3′ to the ORFs of the 12 known yeast HAT catalytic subunits. The addition of auxin to growth medium then induces targeting of the HAT-AID degron fusion proteins for rapid proteasomal degradation (Fig. 4a)[32]. To induce fast degradation of each HAT−AID, cells were treated with 500 µM indole-3-acetic acid (IAA) for 1.5 h under inositol starvation (*INO1* induction) conditions. This condition was sufficient to reduce the expression of each HAT−AID protein as observed by Western blot (Supplementary Fig. 2). After incubation, cells were immobilized on ConA-coated microtiter

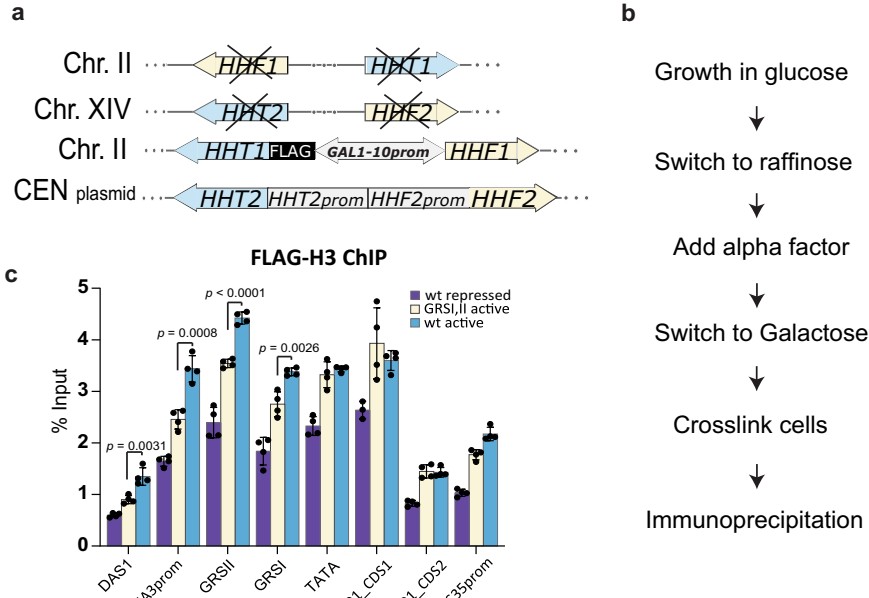

**Fig. 3 | H3 exchange increase in the activated *INO1* is affected when GRS I and II are deleted. a** Schematic representation of the strain used to measure the incorporation of newly synthesized FLAG-H3 into nucleosomes[28]. **b** Cell sample preparation workflow before ChIP. For the repressed condition, inositol was added in all the media used. **c** FLAG-H3 enrichment, as determined by ChIP-qPCR and expressed in percentage of input, in the vicinity of GRS I and GRS II regions within *DAS1, SNA3, INO1 and VPS35* gene loci for the *wild type* strain under repressed (purple) and activated (blue) conditions, and the GRSI, IIΔ (light yellow) under activated conditions. Source data are provided as a Source Data file. Data in panel **c** represent mean ± SD; $n = 4$ biological samples. Data were analyzed by unpaired two-sided Student's *t* test. *p*-values are shown for the regions with significant differences between the GRS double mutant and the *wild type* active gene.

plate wells and imaged to generate high-resolution probabilistic maps of the localization of the *INO1* locus.

Among the 12 HAT catalytic subunits, depletion of three of these reduced partitioning of the active *INO1* locus towards the NE: Gcn5, Rtt109, and Taf1 (Fig. 4b–g, Supplementary Figs. 3 and 4). Deletion of *GCN5*, coding for a subunit of the SAGA complex, has previously been shown to prevent active *INO1* locus partitioning towards the NE[11]. The SAGA complex has also been implicated directly in the activation of *INO1* through transcription factors that bind to the upstream activating sequence (UAS) sites close to the TATA box of *INO1*[33]. Rtt109 is not part of any known chromatin remodeling complex and has not been implicated in *INO1* gene activation previously. This HAT acetylates lysine 56 on the histone H3 core domain (H3K56), a modification that has been globally observed on newly synthesized histones in *S. cerevisiae* and *S. pombe*[28–31,34]. Interestingly, H3K56 acetylation also correlates with higher histone exchange rates in global analyses of the *S. cerevisiae* genome[28]. Finally, Taf1 is a subunit of the large TFIID complex, a general transcription initiation factor for RNA polymerase II (Pol II). Taf1 has been shown to have histone acetyltransferase activity in in vitro assays, but its role in vivo is less well understood[35,36].

We analyzed the effect of the depletion of each of these three HATs on a specific acetylation modification in the regions surrounding the *INO1* locus including the GRS sites. The H3K14 acetylation is known to be catalyzed by Gcn5[37–40], and it has been observed in vitro that Taf1 also acetylates H3 K14[35], although this is more controversial[36]. Rtt109 has been also observed to acetylate H3K14 in in vitro assays[41,42]. This modification was quantified by ChIP-qPCR on the *INO1* locus and on proximal genes in the AID strains for each of the three HATs. Results are presented as the ratio between each modification and H3 levels in each region analyzed. The regions screened correspond to the same ones used when measuring the rate of incorporation of newly synthesized histones (Fig. 3).

The results obtained (Fig. 4h), show that there is a significant decrease in the H3K14ac/H3 ratio for the strain where Gcn5 was

depleted compared to *wild type* in all regions analyzed. A significant decrease is also observed for the other two HATs, mainly for the regions close to the GRSII site, however not as significant as that for the Gcn5-AID strain. This result indicates that Gcn5 could be the dominant HAT for acetylation in these regions.

We quantified mRNA levels for these loci after depleting each of the HATs in the AID strains. We observed a decrease in *INO1* mRNA levels for Gcn5-AID and Rtt109-AID depletion strains, and a decrease for *SNA3* mRNA levels, in this case for all three HAT-AID strains, although more significantly for Gcn5-AID and Rtt109-AID strains (Fig. 4i). For the *DAS1* and *VPS35* loci, there was no apparent difference in mRNA levels between all four strains (Fig. 4i). These results indicate that the transcription is affected for *INO1* and the GRS containing upstream locus *SNA3*, when at least two of the three HATs are depleted. Previous results have shown that deleting the GRS sites cause a decrease in *INO1* mRNA levels[11].

To probe for HATs and other chromatin remodeling complexes proximal to specific GRSs, we developed a proximity-dependent biotinylation (BioID) approach, based on guide RNA (sgRNA)-directed catalytically dead (d)Cas9 (Fig. 5a)[43]. We created BirA*-dCas9 ligase fusions in different N- or C-terminal orientations and using short (12) or long (38) amino acid linkers. These constructions were used to provide for more or less steric constraint and flexibility and promiscuity of biotin labeling. The short linker was used in a previous BioID study in yeast (GSSGSSGSSGSS)[43] and the long sequence in a CasID study in mammalian cells (LERPPLCWISAEFHHTGLVDPSSVPSLSLNRGSGSGSG)[44].

We transformed a *wild type* strain with the different plasmids encoding the BirA*-dCas9 constructs and four different sgRNAs targeting genomic sequences proximal to the GRS I site. We also generated control strains that were transformed with the BirA*-dCas9 constructs but not with sgRNAs (Supplementary Tables 1–3). Western blot confirmed the expression of the different fusion proteins generated (Supplementary Fig. 5). Biotinylated proteins were pulled down on streptavidin-sepharose beads, bound proteins were trypsin digested, and resulting peptides were analyzed by tandem mass

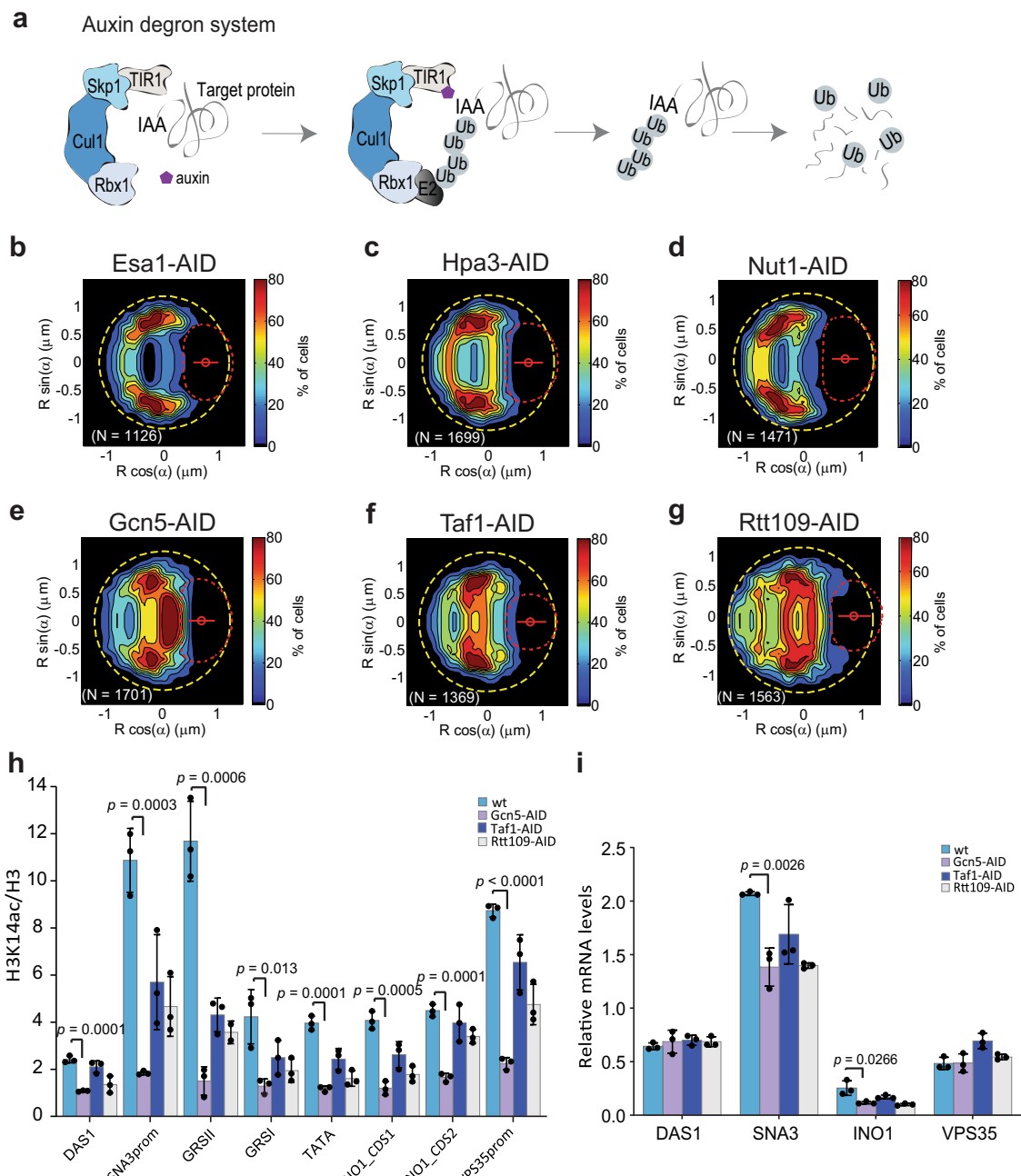

**Fig. 4 | Specific histone acetyltransferases (HATs) are required for active *INO1* localization. a** Schematic representation of the auxin-inducible degron (AID) system used to deplete targeted HAT catalytic subunits for degradation in the presence of auxin. **b**–**g** *INO1* probability maps obtained from the analysis of thousands of nuclei in strains with different HATs tagged with the AID degron and grown in the absence of inositol (active). **b**–**d** HAT-AID tagged strains where there is no effect on the statistical distribution of the *INO1* locus after treating the cells with auxin. **e**–**g** HAT-AID tagged strains where the addition of auxin results in disruption of the *INO1* locus relocalization pattern to the nuclear periphery. **h** H3K14ac enrichment around and within the active *INO1* gene, as determined by

ChIP-qPCR and expressed as H3K14ac/H3 ratio, in the vicinity of *DAS1, SNA3, INO1 and VPS35* gene loci for *wild type* and three different HAT-AID strains treated with auxin and under inositol activating conditions. **i** mRNA levels as quantified by RT-qPCR, relative to *ACT1*, for *DAS1, SNA3, INO1* and *VPS35* loci in *wild type* and Gcn5-AID, Taf1-AID, and Rtt109-AID strains in the presence of auxin and *INO1* activating conditions. Source data are provided as a Source Data file. Data represent mean ± SD (**h** and **i**); *n* = 3 biological samples. Data were analyzed by unpaired two-sided Student's *t* test and *p*-values are shown for significant differences between *wild type* (blue) and Gcn5-AID mutant (light purple). See also Supplementary Figs. 2–4.

spectrometry (LC-MS/MS). We observed enrichment of peptides derived from proteins associated with several nuclear complexes in extracts from strains expressing versus those not expressing sgRNAs (Fig. 5b–e, Supplementary Fig. 6) (Table 1; Supplementary Data 1–6). Five of the complexes are associated with chromatin remodeling activity including Ino80, Swr1, ASTRA, Set3, and FACT complexes (Table 1). Among the enriched peptides, those belonging to the

ATP-dependent DNA helicases Rvb1 and/or Rvb2 were enriched in all separate sgRNA-BirA*-dCas9 samples. Rvb1 and Rvb2 are components of Ino80[45], Swr1[46], and ASTRA[46] complexes and they are also linked to two (Gcn5 and Taf1)[47–49] of the three HAT catalytic subunits, whose depletion resulted in reduced partitioning of the active *INO1* towards the NE. Specifically, Gcn5 and Taf1 make protein-protein interactions with Rvb1 (Gcn5) and Rbv2 (Gcn5 and Taf1) and Taf1 also interacts with

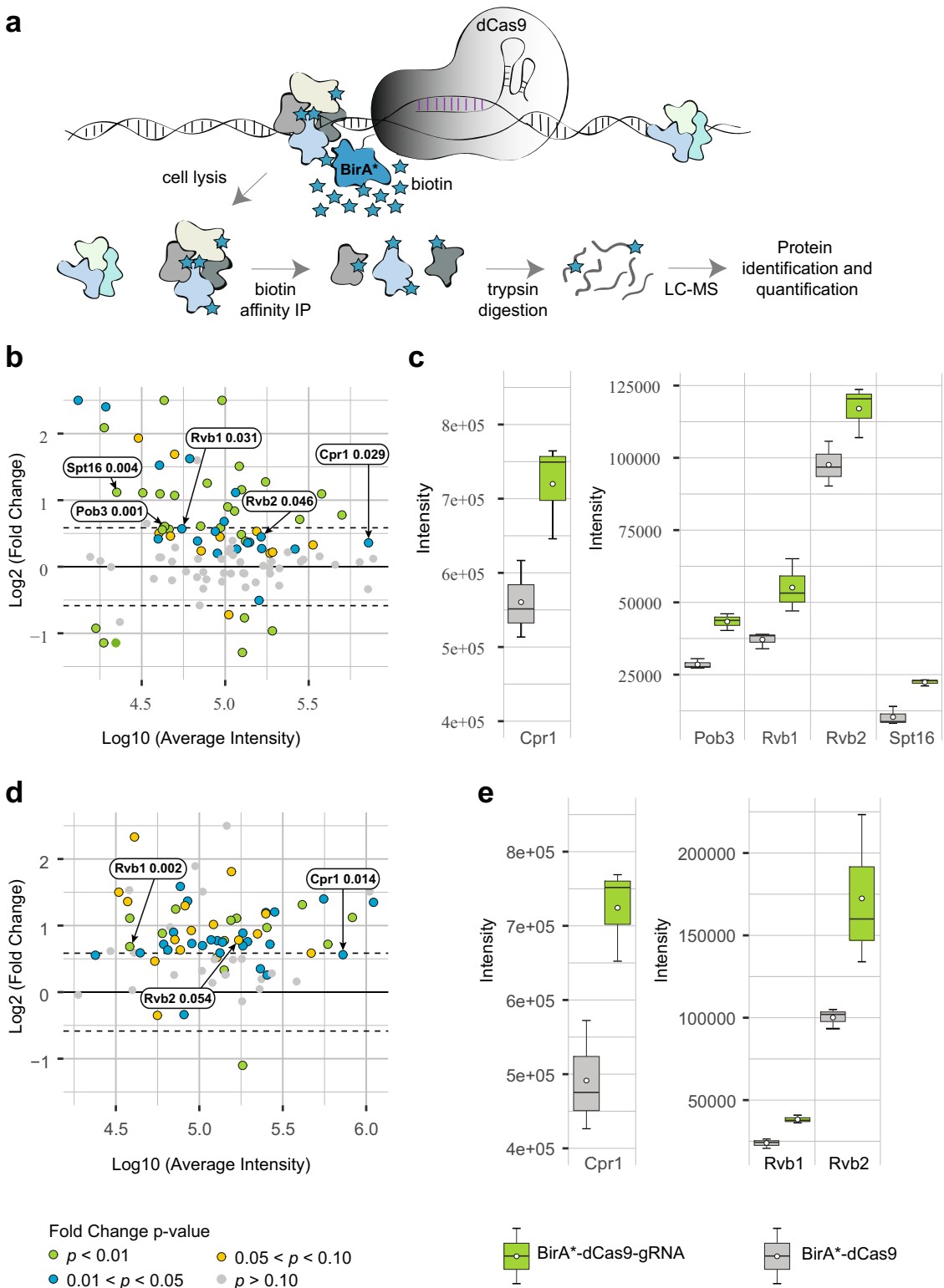

**Fig. 5 | Chromatin remodeling subunits bind GRS site near *INO1* locus. a** BirA*-dCas9 approach designed to target protospacer adjacent motif (PAM) sites in the vicinity of GRS I site by using different sgRNAs for site-specific binding of BirA*-dCas9 and biotinylation of proximal proteins. Proteins were then affinity purified with streptavidin, digested, and analyzed by mass spectrometry. **b**, **d** Micro array (MA) plots representing average protein intensities from biological triplicates from sgRNA-BirA*-dCas9 samples S5 (**b**) and S7 (**c**) plotted against protein quantity fold changes from these sgRNA-BirA*-dCas9 samples over those from their respective negative control triplicates (BirA*-dCas9). Protein fold change *p*-values calculated from a two-sided unpaired Student's *t* test with equal variance are color coded, and the gray dashed lines represent 1.5-fold quantity increase. *p*-values are indicated next to labels. **c**, **e** Box plots of measured protein intensities of candidates from sgRNA-BirA*-dCas9 samples S5 (**d**) and S7 (**e**) triplicates along with their respective negative control triplicates. Source data are provided as a Source Data file. Median is indicated as middle line, average as a white dot, 25th and 75th percentile as boxes and whiskers represent 5th and 95th percentile. False discovery rates thresholds for mass spectrometry identification of peptide spectrum match, peptides, and proteins were set to 0.1. See also Supplementary Data 1–6 and Supplementary Figs. 5 and 6.

**Table 1 | Enriched biotinylated proteins associated with chromatin remodeling and Histone Acetyltransferase (HAT) complexes**

| Enriched biotinylated proteins | Associated chromatin remodeling complexes | HATs that bind to enriched proteins[a] |
|---|---|---|
| Rvb1, Rvb2 | ASTRA complex | Gcn5, Nut1, Sas2, Taf1 |
| | Ino80 complex | |
| | R2TP complex | |
| | Swr1 complex | |
| Spt16, Pob3 | FACT Complex | Elp3, Esa1, Gcn5, Hat1, Rtt109, Sas3, Sua7 |
| Acs2 | histone acetylase complexes | Elp3, Esa1, Gcn5, Hat1, Sas2, Sas3 |
| Cpr1 | Set3 complex histone deacetylase complexes | Nut1, Taf1 |

[a]Reported in the BioGRID DataBase.

Ino80, another subunit of the Ino80 complex. Both subunits of the FACT complex Spt16 and Pob3 showed peptides enriched in one of the sgRNA-BirA*-dCas9 samples, and there are reported interactions of these proteins with several HATs including Gcn5[47,50] and Rtt109[51]. Analysis of protein enrichment in the samples expressing sgRNAs revealed that some proteins were only enriched in one of the samples, or the enrichment of several proteins varied between samples. This may indicate steric label constraint or hindrance due to sgRNA localization on chromatin, BirA*-dCas9 fusion orientation and linker length (Supplementary Fig. 6, Supplementary Data 4–6).

Taken together, our observations imply a probable role for Gcn5, Taf1, and Rtt109 in initiating the cascade of histone acetylation and other remodeling activities that results in increased histone exchange and decompaction of the *INO1* locus chromatin. That there are likely alternative sites of association of these HATs or other chromatin remodeling proteins is consistent with our observation that the GRSI, IIΔ double deletion only results in a partial decrease in histone H3 exchange at the active *INO1* locus.

**Decompacted chromosome segments can undergo polymer-polymer phase separation**

Our results provide biochemical evidence for the mechanism by which changes in material properties of the activated *INO1* gene locus is initiated[17,19]. How then are these changes in material properties consistent with a phase transition that causes partitioning of the *INO1* locus to the NE? Would a change in properties of a single gene locus towards the middle of a chromosome arm cause such narrow localization of the locus to the NE? Phase separation of segments of polymers or polymer-polymer phase separation (PPPS) have been predicted based on Flory–Huggins theory, in which self-interacting ligand molecules occupy multiple sites throughout the polymer[52]. Phase separation between two segments of the polymer will occur if the density of ligands decreases in one region. The high ligand occupancy region forms a dense phase through the high valency ligand-ligand interactions and the low-density segment forms an expanded, "dilute" phase.

Bead-spring models of chromatin provide the means to explore the physical properties of the chromatin fiber and to test hypotheses about mechanisms responsible for polymer, including DNA phase separation. For instance, as predicted by Flory–Huggins theory[53,54], invoking dynamic chromosome cross-linkers (bound ligand) into a model of the yeast genome recapitulates the segregation and morphology of the nucleolus[55]. The physical basis for partitioning the *INO1* gene locus extends beyond recruitment to the NE. The radius of confinement and mobility of the locus are decreased, indicating a change in the polymeric state that accompanies its migration to the NE. While

we can readily identify transcriptionally active domains relative to DNA sequence, we are just starting to appreciate how gene activity manifests in structural changes at the chromosome level. Active genes can be sequestered into chromosome loops (topologically associated domains) or into membrane-less compartments[56,57].

Although Flory–Huggins theory alone can explain segmental phase separation of a polymer as described above[52], we also wanted to test if we could reproduce the dynamic behaviors of the *INO1* locus observed experimentally and see whether segmental phase separation could occur by including parameters in our bead-spring model that are known to accompany changes in gene expression. One of these parameters is the persistence length of the chromatin fiber. As discussed above, persistence length ($L_p$) is the length scale over which the polymer tends to remain linear and resist bending (expected value of cosine of angle θ from the horizontal at length L over $L_p$ falls off exponentially according to Eq. (2)). For the large fraction of non-transcribed chromatin, persistence length measurements range in the hundreds of nanometers. In the case of transcribing gene loci or regions of reduced histone occupancy, these regions have a much shorter persistence length and therefore can explore a greater number of configurations[21–23].

$$\langle \cos\theta \rangle = e^{-\frac{L}{L_p}} \qquad (2)$$

To simulate the experimental condition of active genes flanked by inactive genes, we tuned the model such that the midsection of beads representing active genes has the lowest bending stiffness, $L_p = 5$ nm, flanked by sections of inactive chromatin with $L_p = 250$ nm. The MSD of the 5 nm $L_p$ segment was considerably reduced relative to the 250 nm $L_p$ segment (Fig. 6a), recapitulating our experimental observations (Fig. 2). Likewise, the increase in $k_s$ and reduced $R_c$ of the 5 nm $L_p$ segment further reflect our experimental results (Fig. 6b, c). The 5 nm $L_p$ chain collapsed into a much smaller area compared to the 250 nm $L_p$ segments, due to the increase in number of entropic states that they can explore. Consequently, the 5 nm $L_p$ region appears spherical compared to the ellipsoid shape of the 250 nm $L_p$ due to their increased randomness, and hence increased shape symmetry. The result is that the 250 nm $L_p$ segments showed increased frequencies and amplitudes of expansion-contraction compared to the 5 nm $L_p$ (Fig. 6d). These behaviors reflect our experimental observations of compression/expansion of the *INO1* locus in repressed versus activated states (Fig. 2k). Finally, simulations of chain dynamics reveal phase separation of the 5 and 250 nm $L_p$ segments modeling the phase separation observed in the analytical model and in our experimental results for the active *INO1* locus (Supplementary Movie 1).

We can reproduce segmental changes in dynamics and phase separation of a model polymer, we cannot explain the dynamics itself, which are known to be driven by active, ATP-dependent processes that remain obscure, but which we model as Brownian dynamics with tuning of temperature[58,59].

## Discussion

Our results for *INO1* locus chromatin can now be put into molecular terms to explain the partitioning of the active gene to the NE. Recent evidence suggests that liquid-liquid phase separation (LLPS) of histones combines with PPPS of chromatin to form condensates, a potentially important mechanism by which the spatiotemporal organization of the genome can be regulated[56,57]. Specifically, acetylation-deacetylation of the histone tails in nucleosomal arrays has been shown to underlie PPPS in which deacetylated histones induce formation of a protein-DNA condensate and acetylation causes its dissolution[56]. We propose that the disruption of histone interactions that occurs due to acetylation in the active *INO1* locus causes it to "dissolve" out of the dense nucleoplasmic chromatin phase into the

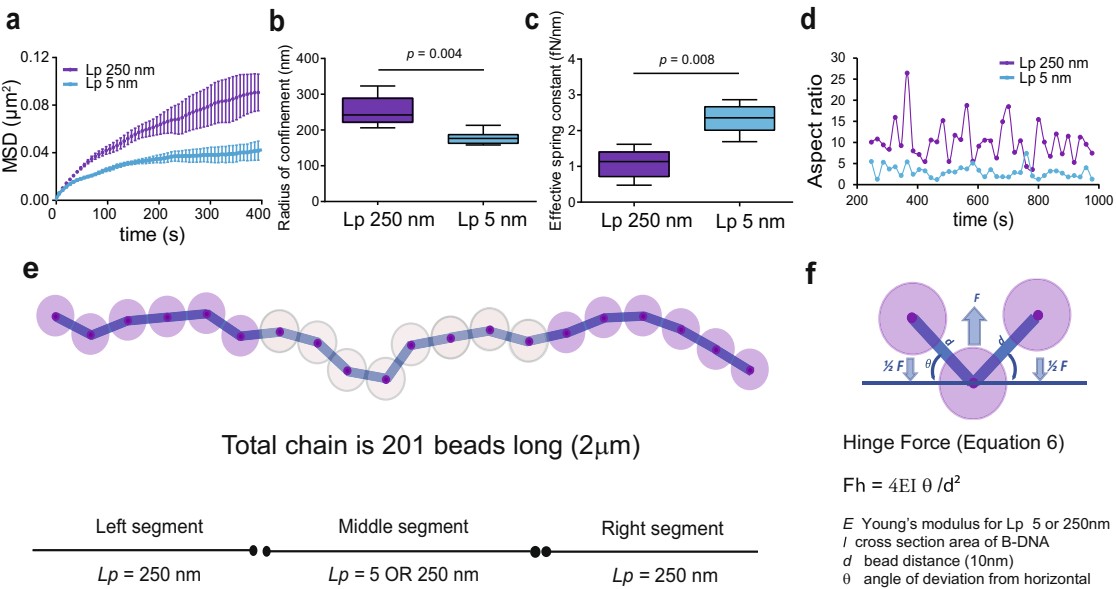

**Fig. 6 | Elastic wormlike chain (bead-spring) modeling recapitulates active ($L_p$ = 5) and repressed ($L_p$ = 250) chromatin material properties. a** Aggregated MSDs for 26-bead subsegment within the high persistence length ($L_p$ = 250 nm) segment represents repressed chromatin (purple, $n$ = 5 randomly seeded simulations), low persistence length ($L_p$ 5 nm) segment represents active chromatin (blue, $n$ = 6 randomly seeded simulations). Data are presented as mean values+/− SEM. Decreasing $L_p$ to 5 nm reduces the radius of confinement (**b**) and increases apparent effective spring constant ($k_s$) for the 26-bead-subsegment by 2-fold (**c**), consistent with experimental observations in Fig. 2b, f. **d** Aspect ratio of modeled fluorescent loci (occupied by the 26-beads) at 38 time points from the simulation averages. Source data are provided as a Source Data file. **e** Cartoon illustration of bead-spring model. At 10 nm rest length, springs experience no force. Beads are soft spheres 9 nm diameter that will resist compression via excluded volume force. **f** Hinge forces applied to beads proportional to varied persistence length will affect bead positions. In the box and whiskers plots (**b**, **c**) the median is indicated as middle line, 25th and 75th percentile as boxes and the whiskers represent minimum and maximum values.

less dense chromatin phase adjacent to the NE, as predicted previously for a single copolymer chain[52].

The increased spring stiffness of the active *INO1* locus we observed in experiments and simulations also have implications for LLPS of proteins associated with sub-compartmentalization of different processes in the nucleus. The contribution of PPPS to formation of chromatin-containing condensates has been considered both theoretically[60] and most recently, demonstrated both in vivo and in vitro[61,62]. Quail et al. recently showed that LLPS of a transcription factor required it to be bound to DNA in a tension-dependent manner. They presented evidence to support the postulate that less dense DNA forms an ideal template for nucleating LLPS of the bound protein and that protein-DNA co-condensation is associated with the difference in chemical potential of the DNA, as reflected in its difference in spring stiffness. Shin et al. observed that light-activated condensates in the nucleus selectively formed in regions of low-density chromatin as observed in vitro[61,62]. Ribosomal DNA (rDNA) is another example of this phenomenon. rDNA is essential for the phase separation of the nucleolus[63]. When rDNA was made discontinuous (rDNA genes were expressed from individual plasmids versus genome repeats), nucleolus formation was disrupted upon transcriptional inactivation[63]. That is, rDNA is a PPPS that relies on transcription to remain phase separated from the bulk genome. Thus, combined protein-DNA LLPS-PPPS of chromatin could facilitate the nucleation of co-condensates that have been associated with transcription and its regulation[64,65], heterochromatin formation[66,67], nucleolus assembly[68] and DNA repair[69].

The work done to sub-compartmentalize the *INO1* and other active gene loci could also be driven by PPPS. Partitioning of the repressed locus into the nucleoplasm phase is driven by transduction of the chemical potential of PPPS into mechanical work on the non-condensed DNA[62]. The work done in sub-compartmentalizing the active *INO1* locus to the nuclear periphery is simply equal and opposite to that driving condensation. It is also possible that the *INO1* chromatin

could nucleate LLPS of associated proteins combined with PPPS, resulting in a DNA-protein condensate.

The polymer modeling provides critical insight into mechanisms of translocation of an active locus. The means by which the subcompartment comes to reside at the nuclear periphery is that it is seeking the most thermodynamically favored state. Chains with self-similar properties tend to segregate as mixing of two chains is thermodynamically disfavored in a confined space, such as the nucleus[55,70]. In the case of the nucleus, segregation of dissimilar chains leads to their migration to the furthest distance from each other, which is the outer edge of the nucleus at the nuclear envelope.

Finally, the ability to partition the genome into discrete domains that transition between physical states, either through multivalent binding of DNA-associated proteins (dense and deformable) or disruption of histone association (dilute but less deformable active subcompartments) could lead to architectural changes that can be understood from the material properties of the components, as we have demonstrated in this study.

## Methods
### Strains and plasmids
The LMY52 strain[12] was used as a control for the statistical mapping and for the gene tracking experiments (provided by Jason Brickner, Northwestern University, USA) (Supplementary Table 1). This strain contains a cassette of 128 repeats of the Lac operator binding site (LacO) into the *INO1* locus and expresses a GFP-Lac repressor (LacI), which binds to the LacO repeats. This strain also expresses the Nuclear Pore Complex (NPC) protein Nup49 fused to GFP and the nucleolar protein Nop1 fused to mCherry as reference markers for the NE and the nucleolus, respectively.

GRS I and II deletion mutants used for the statistical mapping and for the gene tracking experiments were generated in the LMY52 strain by employing a CRISPR-Cas9 based strategy developed by and with reagents provided by Tom Ellis at Imperial College London, UK (Shaw

et al.[71], Supplementary Table 2). This system consists of a sgRNA entry vector and a range of Cas9-sgRNA expression vectors with different selection markers that are transformed into yeast cells together with donor DNA fragments containing the desired sequence changes. In this case, each sgRNA entry vector (pWS082) was completed using small fragment Golden Gate assembly[72] with two 26-mer oligos each containing 20 nucleotide guide sequences selected to target each GRS to be deleted. Once assembled, the plasmid was digested with EcoRV to generate a linear sgRNA expression cassette. This linearized cassette contained homology arms to the Cas9-sgRNA expression vector that served for gap repair of the linearized plasmid in yeast to reconstitute a stable Cas9-sgRNA expression vector. The Cas9-sgRNA expression vector selected for each GRS was also linearized with a BsmBI restriction digest and gel purified. The different markers on each Cas9-sgRNA vector allowed selection for successfully transformed cells expressing both Cas9 under the strong *PGK1* promoter and a sgRNA. Each linear sgRNA expression cassette and the linear Cas9-sgRNA expression vector were transformed into LMY52 together with the corresponding donor sequence consisting of a double stranded DNA sequence of around 100 bp obtained by synthesizing and annealing mutagenic primers containing the corresponding GRS mutated sites flanked by homology arms to the integration region at the *INO1* locus.

The YAN1001 strain[28] (Supplementary Table 1) was used as control to measure the incorporation of newly synthesized histone H3 into nucleosomes (provided by Amine Nourani, Université Laval, Canada). In this strain, endogenous genes for histones H3 and H4 are deleted and expressed in two different ways: 1, constitutively from a plasmid, under the control of their own promoter, and 2, from another region in the genome under the control of the *GAL1-10* promoter, with H3 fused to a FLAG tag. This allows to measure incorporation of newly synthesized H3-FLAG into nucleosomes following galactose induction. GRS I and II (GRSI, IIΔ) deletion mutants (Supplementary Table 1) were generated in the YAN1001 by the CRISPR-Cas9 strategy described above.

To measure the effect of rapid depletion of different HATs on active *INO1* localization to the NE, 12 known yeast HAT catalytic subunits were tagged with a degron peptide IAA17 (71/116) that targets the protein for proteasomal degradation in the presence of the plant hormone auxin[32]. First, the *TIR1* gene that mediates the auxin-IAA degradation was integrated into the genome of the LMY52 strain by transformation with the Yeast Integrating plasmid YIp204-PADH1-atTIR1-9myc (Addgene #99532, Supplementary Table 2). This strain was used as background to tag each HAT 3' to the coding sequence to produce C-terminal fusion products with a cassette containing the IAA17 degron sequence as well as six FLAG epitopes amplified from a plasmid generated in a previous study (pHyg–AID*–6FLAG, Supplementary Table 2)[32]. These FLAG tags allowed us to verify the efficiency of degradation by Western Blot.

BirA*-dCas9 fusions to identify candidate GRS-binding proteins and specifically, histone acetyl transferases (HATs), were generated based on a previously described Proximity-dependent Biotin Identification (BioID) strategy that was successful in *S. cerevisiae*[43]. The BirA* ligase sequence was provided in a plasmid vector by Oliver Valerius at Georg-August-University, Germany (Supplementary Table 2). dCas9 mutant was first obtained in one of the Cas9-sgRNA expression vectors (pWS174) to introduce the mutations D10A and H840A resulting in a catalytically dead (d)Cas9. Then, by linearizing this vector at the selected fusion site and using the GeneArt Seamless Cloning and Assembly kit (ThermoFisher Scientific), the sequence encoding for BirA* with specific linker sequences (linker 1: GSSGSSGSSGSS or linker 2: LERPPLCWISAEFHHTGLVDPSSVPSLSLNRGSGSGSG) were fused 5' or 3' to the dCas9 gene to produce N-terminal or C-terminal fusion products, respectively. Once we obtained the desired BirA*-dCas9 expression vectors, the same procedure was performed as described above for CRISPR/Cas9 genome editing by transforming both

BirA*-dCas9 and sgRNA linearized cassettes into the LMY52 strain to obtain the strains used in this approach (Supplementary Table 1), this time without the need of using any donor DNA sequence. We also transformed the LMY52 strain with the different BirA*-dCas9 expression vectors alone with empty sgRNA cassettes as negative controls for non-targeted biotinylation of proteins.

## Statistical mapping of the *INO1* locus

Statistical mapping of the *INO1* locus was performed as previously described[12]. Briefly, overnight cultures in synthetic, defined medium (SDC) without inositol or SDC containing 100 μM of inositol were diluted to $10^6$ cells/mL and after two generations cells were immobilized on Concanavalin A-coated well slides[73]. Images were acquired with a Zeiss Axio Observer Z1 Yokogawa spinning disk confocal microscope using a 100× 1.40 NA oil objective. Z-stacks with a slice spacing of 250 nm were taken at an acquisition time of 50 ms for GFP (50%, 3 mW 488 nm excitation), and 100 ms for mCherry (50%, 3.3 mW 561 nm excitation). Images were processed by nucloc software modified to display probability density maps as percentiles using a kernel density estimate[74].

## Particle tracking of the *INO1* locus

To track the *INO1* locus in live yeast cells, LMY52 and GRS I or II mutant strains containing the gene tagged with the LacO array and expressing GFP-LacI as well as the NE tagged with Nup49-GFP were used. Images of the cells were also recorded on a Zeiss spinning disk Axio Observer Z1 confocal microscope at an acquisition time of 50 ms (50%, 3 mW 488 nm excitation) for 1 min at an interval of 500 ms. To check for possible stage-drift the GFP signal was also imaged in fixed cells.

Images of GFP foci corresponding to the *INO1* locus were automatically tracked using the WaveTracer tool in MetaMorph software[75]. The algorithm segments circular objects with a particular size and intensity cut-off and then calculates the centroid position for each object in each frame through Gaussian fitting. For each locus monitored, the software quantifies the movement and calculates mean squared displacement (MSD). Only cells whose MSD curves exhibited a linear slope were included in subsequent analysis.

To calculate the radius of confinement, the MSD plateau values were used in each case since the plateau reached is proportional to the square of $R_c$, as previously described (Eq. (3))[17,25].

$$R_c = \frac{5}{4}\sqrt{MSD_{plateau}} \tag{3}$$

or independently, from the standard deviation of spot positions, σ, and the average squared deviation from the mean position, $\langle \Delta r_0^2 \rangle$, by applying the equipartition theorem (Eq. (4))[17,26,27].

$$R_c = \frac{5}{4}\sqrt{2\sigma^2 + \langle \Delta r_0^2 \rangle} \tag{4}$$

Effective spring constants ($k_s$) for the *INO1* locus in repressed versus activated conditions were determined by using the equipartition theorem to measure the standard deviation (σ) of each step from the mean position to calculate $k_s$ (Eq. (5))[17,26].

$$k_s = \frac{k_B T}{\sigma^2} \tag{5}$$

Statistical differences for $R_c$ and effective $k_s$ in strains grown under different conditions were determined by Student's *t* tests.

To analyze the rate and amplitude of *INO1* locus expansion-contraction, the GFP array was also acquired at the same frequency as mentioned before but using a Zeiss Elyra PS.1 system. SIM images were acquired in this case with a 63× 1.40 NA oil objective in the GFP channel

(15 %, 20 mW 488 nm HR diode laser), exposure time 24 ms and 3 rotations. Each locus was then fitted to an ellipsoid function to quantify the ratio between long and short axes by using the Fiji software[76].

## Chromatin polymer simulations

An elastic worm-like chain consisting of 201 beads (masses) connected by linear springs was sectored into three equal segments: two flanking segments composed of beads 1–67 and 135–201, and one middle segment with beads 68–134. Initially the chain was homogenized with $L_p$ for all segments set to 250 nm (in $n = 5$ randomly seeded simulations). In the experimental condition we lowered the $L_p$ for the middle segment to 5 nm (in $n = 6$, randomly seeded simulations) by adjusting the value of hinge force on the beads. Whenever three beads (or two springs) are not colinear, hinge forces act to restore the positions of the beads to lie on a line (see Fig. 6e, f). Hinge forces are based on the Euler–Bernoulli formula (Eq. (6)), where $E$ is Young's modulus, proportional to a polymer's bending stiffness, and is linearly related to the persistence length (for more details see methods Lawrimore et al.[77]). In our approach the Young's modulus (E) in Eq. (6) is varied only in calculation of bending stiffness, and does not affect the tensile stiffness of individual DNA springs. Hinge forces were manually set to constant $L_p = 250$ nm for the flanking segments, and set to either $L_p = 250$ nm and $L_p = 5$ nm for the variable middle segment.

$$F_h = \frac{4EI\theta}{d^2} \tag{6}$$

In the simulations we obtained 35,000 timepoints of bead position data ($x_t$, $y_t$, $z_t$) for a duration of 70 ms at 2 μsec resolution. With a working estimate of cellular nuclear viscosity of 141 Pa[78] the model's parameters reflected a fluid environment set to 0.01 Pa allowing a 14,100X speedup of physical time for the simulation. An equilibration time corresponding to 25% of the total trajectories was set to allow for meaningful chain configurations from its starting linear conformation. Second, to eliminate the effects of drift on the entire chain, we considered individual bead positions relative to bead number 55 within the invariable portion of the chain ($L_p$ 250 nm segments) throughout our analyses. Positions of a center of mass of a 26-bead long subsegment (beads 90–115) within the middle segment representing a fluorescent segment (virtual LacO array segment), were tracked for each of 15,000 time intervals (t values) amounting to 35 ms of simulation time (equivalent to 423 s of real time), and MSD was computed for each independent experiment in MATLAB with *computespotmsd.m* (Eq. (7)). Average MSD with standard error bars for a spot (26-bead subsegment) within $L_p$ 250 ($n = 5$) compared to spot within $L_p$ 5 nm ($n = 6$) chain segments are displayed in Fig. 6a.

$$MSD = \langle |x_t - x_o|^2 \rangle \tag{7}$$

We computed $R_c$, for the simulated with Eq. (4)[17].

$$R_c = \frac{5}{4}\sqrt{2\sigma^2 + \langle \Delta r_0^2 \rangle} \tag{8}$$

Considering 35 ms of simulation time and using the standard deviation of spot position, $\sigma$, and the averaged square deviations from the mean position, $\langle \Delta r_0^2 \rangle$ we compared spatial confinements of $L_p$ 250 nm ($n = 5$) to $L_p$ 5 nm segments ($n = 6$). Effective spring constants were estimated by using the standard deviations obtained from simulation's spot positions, σ, as in Eq. (5).

$$k_s = \frac{k_B T}{\sigma^2} \tag{9}$$

We used principal component analysis to estimate 2D shapes of simulated fluorescent spots (26-beads) to obtain their aspect ratios. Given averaged bead position data at 38 regularly spaced time intervals we used MATLAB *aspect_ratio.m* to calculate the aspect ratio for the point cloud representing the fluorescent spot (26-bead subsegment) by taking the square root of the ratio of the first principal component's scaling factor over the second.

## ChIP experiments

**Cell cultures and crosslinking.** ChIP experiments were performed in four biological replicates. YAN1001 or the corresponding GRS mutant strains were inoculated on 100 mL SDC + 2% glucose with or without 100 μM inositol medium and incubated until they reached OD$_{600}$ 0.2–0.3. Cells were recovered by centrifugation and switched to 100 mL medium with 2% raffinose and incubated overnight. In the morning, cell cycle was arrested with 5 mM α-factor for 3.5 h and inositol was added again to 100 μM inositol final concentration to the repressed cells. Efficient G1 arrest (at least 90% of cells) was confirmed by microscope as percentage of cells with shmoo morphology. Cells were collected by centrifugation at 2000 × g, 5 min switched to 2% galactose medium and incubated for 1 h. At every step, OD$_{600}$ cells were grown only long enough to reach values of <0.5, to avoid losing the repressed phenotype.

After 1 hour incubation in galactose, cells were crosslinked with 1 % formaldehyde for 30 min with gentle rotation at room temperature. To quench formaldehyde, 2.5 M glycine was added to a final concentration of 125 mM. Crosslinked cells were recovered by centrifugation and washed twice with 25 mL ice cold 1X Tris-buffered saline (TBS: 20 mM Tris, 150 mM NaCl, pH 7.6). Pellets were resuspended in the remaining liquid and transferred to 1.5 mL tubes. Cells were harvested by centrifugation at 2000 × g, 4 °C, 5 min, immediately frozen in liquid nitrogen and stored at −80 °C until processed.

**Chromatin immunoprecipitation (ChIP) and quantitative PCR (qPCR).** Immunoprecipitation assays were performed as previously described[79] with slight modifications. Cells were resuspended in 700 μL of lysis buffer (50 mM HEPES-KOH pH 7.5, 140 mM NaCl, 1 mM EDTA, 1% Triton X-100, 0.1% Na-deoxycholate, 1 mM PMSF, 1 mM Benzamidine, 10 μg/mL Aprotinin, 1 μg/mL Leupeptin, 1 μg/mL Pepstatin) and mechanically lysed with glass bead beating. Cells were spun at 5000 × g for 10 s and supernatants were transferred to new tubes. Cells were sonicated in a Sonic dismembrator Model 100 equipped with a micro probe (Fisher Scientific), 4 × 20 s at output of 7 Watts, with a 1-minute break between sonication cycles. Supernatants containing solubilized chromatin fragments were recovered by centrifugation at 16,000 × g, 5 min and transferred to new tubes. An aliquot of 5 μL (1 %) per sample of this whole-cell extract was kept at −20 °C to be used as input sample, which represents the amount of chromatin used in the ChIP, usually 1% of starting chromatin. 500 μL of the remaining material was used for immunoprecipitation (IP). For each IP sample, 50 μL of Dynabeads™ Pan Mouse IgG (ThermoFisher Scientific) pre-coupled to 5 μg of mouse Monoclonal Anti-FLAG M2 antibody (Sigma) was added to the 500 μL of chromatin sample and incubated overnight at 4 °C.

Beads were washed twice with Lysis buffer, twice with Lysis buffer 500 (Lysis buffer + 360 mM NaCl), twice with Wash buffer (10 mM Tris-HCl pH 8.0, 250 mM LiCl, 0.5% NP40, 0.5% Na-deoxycholate, 1 mM EDTA) and once with TE (10 mM Tris-HCl pH 8.0, 1 mM EDTA). Immunoprecipitated chromatin was eluted and reverse-crosslinked with 50 μL TE/SDS (TE + 1% SDS) by incubating overnight at 65 °C. Input samples were also reverse-crosslinked with 50 μL TE/SDS and incubated overnight at 65 °C. Eluted chromatin as well as Input samples were treated with RNase A (345 μL TE, 3 μL 10 mg/mL RNase A, 2 μL 20 mg/mL Glycogen) at 37 °C, for 2 h and subsequently digested with Proteinase K (15 μL 10% SDS, 7.5 μL 20 mg/mL Proteinase K) at 37 °C for 2 h. Samples were extracted twice with phenol/chloroform/

isoamyl alcohol (25:24:1) and precipitated with 200 mM NaCl and 70% EtOH. Precipitated DNA was resuspended in 50 μL of TE before being used for ChIP-qPCR experiments.

The immunoprecipitated DNA and DNA from 1% of the Input samples were analyzed by quantitative PCR on a ViiA 7 Real-Time PCR System (Thermo Fisher Scientific) using the PowerUp™ SYBR™ Green Master (Thermo Fisher Scientific). The amount of newly synthesized FLAG-H3 incorporated at the nucleosomes was calculated as the percent of Input using the following formula: $100*2\hat{}(Ct_{Input}-6.644-Ct_{IP})$, Ct being the value where the PCR curve crosses the threshold line above background levels of detection. Subtracting 6.644 from $Ct_{Input}$ allows to adjust for the fact that 1 % of Input was used. In this case since the starting input fraction is 1%, then a dilution factor of 100 or 6.644 cycles ($\log_2$ of 100) is subtracted from the Ct value of diluted input. Primer sequences for the regions analyzed are described in Supplementary Table 4.

### HATs depletion using an auxin degron system

All 12 HAT-AID fusion strains (Supplementary Table 1) were generated to rapidly deplete specific HAT activity and the *wild type* cells were cultured in SDC medium without inositol until $OD_{600}$ 0.3. At this point, cultures were split in two and auxin (I2886, Sigma-Aldrich) was added to one of the tubes for each sample to a final concentration of 500 μM. Cells were incubated with auxin for 1 h 30 min followed by immobilization on ConA-coated Nunc 96 well Optical-Bottom plates (164588, ThermoFisher). Imaging and nucloc analysis were performed as previously described[12] for the Statistical Mapping of the *INO1* locus *wild type* and the GRS mutants. Western Blots were performed on HAT-AID fusion strains to verify the degradation of the corresponding HAT once the auxin was added to growth medium.

ChIP-qPCR experiments for H3K14ac and H3 in the AID strains were performed in three biological replicates similarly as described before for the H3-FLAG with modifications for the sample preparation and the IP procedures. In this case, cells were grown under the same conditions as described for the AID experiments before crosslinking. For the IP protocols, 50 μL of Dynabeads™ Protein G beads (Thermo-Fisher Scientific) pre-coupled to 2 μL of rabbit anti-H3K14ac (Millipore) or 2 μg of rabbit anti-H3 (Abcam) were used per IP sample.

### Reverse transcriptase real-time quantitative PCR (RT-qPCR)

Cells were prepared under the same conditions described before for AID and inositol starvation. Total RNA was isolated from yeast cells using TRIzol reagent (Invitrogen) following manufacturer's instructions. Approximately 2 μg of total RNA was reverse transcribed using All-In-One 5X RT MasterMix (abm). qPCR reactions were performed using FastStart Essential DNA Green Master (Roche Diagnostics) and gene expression levels for *INO1* and proximal loci (Supplementary Table 4) were determined with LightCycler 96 (software version 1.1). All assays were performed in biological triplicates.

### dCas9-BioID assays

**Cell cultures and streptavidin affinity purification.** The protocol for biotinylated proteins purification was modified from Opitz et al. [43]. LMY52 *wild type* and LMY52-BirA*-dCas9 with or without sgRNA mutants generated (Supplementary Table 3) were inoculated in 4 L of SDC medium without inositol plus D-biotin (B4639, Sigma-Aldrich) added to final concentration of 10 μM. Cells were grown to $OD_{600}$ of 0.8 and harvested by filtration using a Kontes™ Ultra-Ware™ Micro-filtration Assembly with Fritted Glass Support (90 mm) and MF-Milli-pore™ Membrane Filters, 0.45 μm pore size (HAWP0900 Millipore). Pellets were immediately frozen in liquid nitrogen and kept at −80 °C. Cell lysis was performed under cryogenic conditions using solid phase milling in a planetary ball mill (Retsch PM 100) producing a fine cell grindate[80]. Grindate samples were stored at −80 °C until processed. Biotinylated proteins were purified using streptavidin-sepharose

beads (GE Healthcare, 17-5113-01). For each sample, 0.7 g of crude extract was resuspended in 6.3 mL of buffer A: commercial buffer W (100 mM Tris-Cl, 150 M NaCl, pH 8, 2-1003-100, IBA) containing 0.1 % SDS, 0.5 % Sodium Deoxycholate, and protease inhibitors mixture (cOmplete™, Mini, EDTA-free Protease Inhibitor Cocktail, 11836170001, Roche). Samples were homogenized with a vortex agitator, polytron (PT1200E, for 30 s, one time), and incubated at 65 °C for 5 min in a water bath. Subsequent clarification was performed by two rounds of centrifugation at 2465 × *g*, 4 °C, 5 min, and 20,292 × *g*, 4 °C, 20 min, respectively. Triplicates for each sample were prepared; clear extracts were incubated with 30 μL of beads previously equilibrated with buffer A. Samples with beads were incubated for 3 hours at 4 °C with gentle mixing (VWR® Nutating Mixer). Beads were washed with buffer A and transferred to 1.5 mL Protein LoBind tubes (Eppendorf, 0030108422) and three consecutive washes with buffer A were performed, followed by four quick washes with 50 mM Ammonium Bicarbonate pH 8 (ABC). 30 μL of 50 mM ABC pH 8 containing 6 mM fresh DTT were added to each tube followed by 30 min incubation at 37 °C with gentle agitation in a ThermoMixer (Eppendorf) at 350 rpm. Samples were spun quickly and 5μL IAA 126 mM dissolved in ABC 50 mM pH 8 were added to each tube and incubated for another 30 min in the dark at room temperature. Tubes were quickly spun again and 5 μL Trypsin/LysC Mix (Promega, V5073) dissolved in ABC 50 mM pH 8 at 200 ng/μL were added to each one. Samples were digested on-beads overnight at 37 °C, 350 rpm. The next day, after a quick spin, 60 μL of MS grade $H_2O$ were added to the beads, mixed, and centrifuged at low speed (1 min, 400 × *g*). Eighty (80) μL of the peptides digests were transferred to new labeled Protein LoBind tubes before adding formic acid (FA) to a final concentration of 4% v/v. The peptides were dried in a SpeedVac at room temperature and then stored at −80 °C.

**Mass spectrometry and data analysis.** Samples were resuspended in 11 μL of 2 % acetonitrile (ACN) and 1 % FA containing 0.5X of iRT standard peptides (Biognosys, Ki-3002-1), of which, 2 μL of each sample was further diluted 1/5 in the same buffer. Diluted samples were loaded (2.5 μL each) at 450 nL/min on a 17 cm × 75 μm i.d. PicoFrit fused silica capillary column (New Objective), packed in-house with Jupiter 5 μm C18 300 Å (Phenomenex). The column was mounted in an Easy-nLC II system (Proxeon Biosystems) and coupled to an Orbitrap Fusion mass spectrometer (ThermoFisher Scientific) equipped with a Nanospray Flex Ion source (Proxeon Biosystems). Peptides were eluted at a flow rate of 250 nL/min on 2-slope gradient made with 0.2% FA in water (buffer A) and 0.2% FA in 100% ACN (buffer B). Concentration of buffer B first increased from 2% to 36% over 105 minutes, and from 36 % to 80 % over 12 minutes. The mass spectrometer was operated in data-independent acquisition mode. Full MS scans in the range of 400-1000 m/z range were acquired in the Orbitrap at a resolution of 120 K. Each full scan was followed by 25 MS2 acquisition windows with 24 m/z increments and 0.5 m/z overlaps, covering each full scan. Matching MS2 m/z window precursor ions were fragmented by HCD at a 30% collision energy and acquired in the Orbitrap at a 30 K resolution. Both MS and MS2 AGC target were set to 4e5 with maximum fill times of 60 ms. The lock mass option (lock mass: m/z 371.101233) was used for internal calibration. MS RAW files were analyzed with Spectronaut v15.1.210713 (Biognosys) by directDIA. The Fasta database was downloaded from Uniprot and consisted of all *Saccharomyces cerevisiae* S288c verified nuclear proteins entries (1749). Considered peptide length for Pulsar search was 6-52 amino acids with two missed cleavages tolerance, using the semi-specific Trypsin/P rules. Carbamido-methylation of cysteines was set as fixed modifications while oxidation of methionines, protein N-termini acetylation, biotinylation of lysines and protein N-termini as well as phosphorylation (STY) were set as variable modifications, with a maximum of five. Peptide spectrum match, peptides, and protein group FDR were set at 0.01. Machine

learning was applied per run and both precursor and protein were identified at a Qvalue cutoff of 0.01. Proteins were quantified at the MS2 level on stripped peptide median quantities. Unique peptides were used for quantification. Only proteins for which values from all biological triplicates were measured in the sgRNA-BirA*-dCas9 were kept for quantification, and no values were imputed during the analysis. The resulting protein datasets were crosschecked with a list of chromatin and transcription related proteins obtained from Uniprot (Supplementary Data 5). The protein list and associated ontologies was built from searches using the keyword chromatin (461 reviewed entries), and transcription (1211 reviewed entries) in *S. cerevisiae*. After merging both lists and removing duplicates, 1320 entries remained (Supplementary Table 6). A two-sided unpaired Student's t-test with equal variance was used to help in the evaluation of significant protein enrichment from sgRNA-BirA*-dCas9 triplicates. Analyses of evidence for association of chromatin remodeling subunits with nuclear protein complexes was performed for all significant candidates with definitions of subunit compositions of protein complexes taken from the Saccharomyces Genome Database (SGD)[81] and UniProt[82], and protein-protein interactions were identified in The Biological General Repository for Interaction Datasets (BioGrid)[83]. For representation clarity, proteins having a log2 fold change above 2.5 were limited at the 2.5 value on MA plots (Fig. 5). We removed two sgRNA-BirA*-dCas9 samples out of a total of 4 because they did not reveal significant fold changes for many or most of the candidates that were identified in the samples kept for further analysis. Both MA and box plots were made in RStudio (v2022.07.2) using the ggplot2 package (v3.3.6). Data are available via ProteomeXchange with identifier PXD029913.

### Reporting summary

Further information on research design is available in the Nature Portfolio Reporting Summary linked to this article.

## Data availability

Mass spectrometric data are publicly available *via* ProteomeXchange. Project accession: PXD029913. The data supporting the results presented in this work are available in the main article and the supplementary information. Any additional information is available upon request from the corresponding author. Source data are provided with this paper.

## Code availability

All original code for matlab methods specifically referenced in the text are accessible in a GitHub repository publicly available: https://github.com/kolbincode/INO1-paper. https://doi.org/10.5281/zenodo.7492891.

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

## Acknowledgements

The authors thank Jason Brickner (Northwestern University), Tom Ellis (Imperial College London), Amine Nourani (Université Laval), Oliver Valerius (Georg-August-University), and Daniel Zenklusen (Université de Montréal) for providing reagents, plasmids and strains. The authors acknowledge support from Canadian Institutes of Health Research (CIHR) grant MOP-GMX-152556 and Human Frontier Science Program grant RGP0034/2017 (S.W.M.); CIHR grant PJT-153313 (M.O.); CIHR grant PJT-162334 (F.R.); and National Institutes of Health grant R37 GM32238 (K.B.).

## Author contributions

Conceptualization: L.G. and S.W.M.; Methodology: L.G., D.K., C.T., C.J.; Investigation: L.G., D.K., C.T., C.J.; Data analysis: L.G., D.K., C.T., C.J., F.R., M.O., K.B., S.W.M.; Writing-review: L.G., D.K., C.T., C.J., F.R., M.O., K.B., S.W.M.

## Competing interests

The authors declare no competing interests.
