## [Peer Review File · Nature Communications]

Adaptive partitioning of a gene locus to the nuclear envelope in *Saccharomyces cerevisiae* is driven by polymer-polymer phase separationREVIEWER COMMENTS

Reviewer #1 (Remarks to the Author):

In this manuscript, Gonzalez et al. investigate the mechanisms of translocation of the yeast *Saccharomyces* INO1 gene locus to the nuclear envelope (NE) upon activation. Using microscopy experiments and polymer modeling simulations, they highlight a polymer-polymer phase separation of the active locus driven by the increased stiffness of chromatin induced by local histone acetylation. They also provide evidence of the role of GRSI,II, previously shown to be important for the locus partitioning, as histone acetylation association sites. Consistently, depletion of histone acetyltransferases (HATs) by auxin-inducible degron (AID) experiments shows a loss of the INO1 localization at the NE.

I find the manuscript interesting as it contributes spreading light on the mechanisms linking gene activity and chromosome structural changes related to cell function. However, the manuscript clarity and some specific aspects should be improved, as detailed below.

Specific comments:

1) Please correct Figures references across the text, e.g., Figure 2.1 should be Figure 1, Figure 2.2 should be Figure 2, and so on.

We thank the reviewer for pointing this out; we have corrected all figure labels in the manuscript.

2) It would be useful to add a paragraph at the end of the Introduction summarizing the main results of the paper to improve clarity

Thank you for this suggestion. At line 66 we have added the following text:

“Here we present evidence that the GRSs are sites of association of chromatin remodeling complexes, including HATs, which catalyze histone H3 acetylation. Consequently, there is an increase in H3 exchange between nucleosomes and nucleoplasm within and surrounding the activated *INO1* locus. This increase in H3 exchange causes dissociation of nucleosomes. The resulting decompaction of chromatin causes phase separation of the *INO1* locus from the compacted chromatin found in the nucleoplasm to the decompacted chromatin adjacent to the nuclear envelope.”

3) It would be helpful to add a discussion on how the active locus PPPS can mechanistically drive its translocation to the NE, which is unclear to me from the results.

As we write in the Discussion section, there is a remarkable consistency of analytical theory of phase separation of polymers with interacting ligands and recent experimental evidence of histone-acetylation-mediated phase separation of nucleosome arrays *in vitro*. As we state at line 387:

“Specifically, acetylation-deacetylation of the histone tails in nucleosomal arrays has been shown to underlie PPPS in which deacetylated histones induce formation of a protein-DNA condensate and acetylation causes its dissolution¹.”

Like all models, however, the purpose of the modeling is to provide intuition regarding the physical principles that could govern the special dynamics of chromatin in a constrained and viscous environment, one that we otherwise have little intuition for. The model reveals that phase separation is sufficient to drive the translocation of the active PPPS to the nuclear envelope. While it is likely that there are active processes

that contribute to the dynamics of chromatin, these act in random directions and are implicit in our model. The model brings a critical perspective for expecting the active PPPS in this position. We have already partly addressed this point in the Results section, line 339, of the revised manuscript.

“Bead-spring models of chromatin provide the means to explore the physical properties of the chromatin fiber and mechanisms responsible for polymer, including DNA phase separation. As predicted by Flory-Huggins theory^{2,3}, invoking dynamic chromosome cross-linkers (bound ligand) into a model of the yeast genome recapitulates the segregation and morphology of the nucleolus⁴. The physical basis for partitioning the *INO1* gene locus extends beyond recruitment to the NE. The radius of confinement and mobility of the locus are decreased, indicative of a change in the polymeric state that accompanies its migration to the NE. While we can readily identify transcriptionally active domains relative to DNA sequence, we are just starting to appreciate how gene activity manifests in structural changes at the chromosome level. Active genes can be sequestered into chromosome loops (topologically associated domains) or into membrane-less compartments^{1,5}.”

We also added a paragraph at line 423, Discussion section of the revised manuscript:

“The polymer modeling provides critical insight into mechanisms of translocation of the active locus. The means by which the sub-compartment comes to reside at the nuclear periphery is that it is seeking the most thermodynamically favored state. Chains with self-similar properties tend to segregate as mixing of two chains is thermodynamically disfavored in a confined space, such as the nucleus^{4,6}. In the case of the nucleus, segregation of dissimilar chains leads their migration to the furthest distance from each other, which is the outer edge of the nucleus at the nuclear envelope.”

4) The use of the term “stiffness” is confusing in the text because sometimes it is referred to the spring constant of the polymer (which is higher when the persistence length is lower) and sometimes to the bending stiffness of the polymer quantified by the persistence length (higher when the persistence length is higher). I suggest naming them differently, maybe stiffness and bending stiffness, for the sake of clarity.

We thank the reviewer for this comment and have clarified the use of stiffness as recommended. Lines 28, 30, 72, 94, 138, 170, 211, 365, 395, 404, 718, and 720 of the revised manuscript

5) Line 133 should be rephrased since the stiffness is not a measure of the state of compaction.

We have rephrased the sentence at line 138 of the results section:

“We measured differences in the mechanical spring stiffness of the *INO1* locus in the repressed and activated states.”

6) Is the effect on the statistical distribution of *INO1* after HAT-AID (Figure 4 b-d, e-g) been evaluated quantitatively? Please clarify.

Yes, the nucloc software used for the statistical analysis allows plotting cumulative distribution functions for parameters such as the distance to the nuclear envelope for each of the set of images generated. In this way it is possible to quantitate differences in the two-dimensional (2D) localization distributions of genes under different conditions.

In this case, distributions shifted closer to 0 μm indicate that active loci are closer to the nuclear envelope. Additionally, a Kruskal-Wallis test comparing the distance-to-the-NE datasets showed a significant difference ($p < 0.0001$), and further pairwise comparisons

also showed significant differences, all with p values below 0.0001, for each distance-to-the-NE dataset showing a more negative distribution compared to each dataset with a less negative distribution.

A supplementary figure was added to the document:

Supplementary Fig. 4

Cumulative distributions of distances to the nuclear envelope for the active *INO1* locus in six auxin-induced histone deacetylase degradation strains. Distributions shifted closer to 0 μm indicate that active loci are closer to the nuclear envelope. Source data are provided as a Source Data file.

7) Figure 3 panel c is redundant and can be removed since it shows the same plot of panel d, without the GRSI,II active case (black bars in d).

We have removed Fig. 3 panel c as recommended.

Minor/typos:

8) Figure 2b-d caption “Radius of confinement (R_c) significantly increases...”: increases should be decreases

We have corrected the caption for Fig. 2b-d:

“Radius of confinement (R_c) significantly decreases in wild type, $GRS1\Delta$, and $GRS11\Delta$ cells from repressed (red) to activated (blue) conditions.”

9) In Figure 5, caption title: L_p 5 should be $L_p = 5$.

The caption title for Fig. 5 (now Fig. 6) was corrected:

“Elastic wormlike chain (bead-spring) modeling recapitulates active ($L_p = 5$) and repressed ($L_p = 250$) chromatin.”

Reviewer #2 (Remarks to the Author):

González et al. in their manuscript NCOMMS-22-20484 entitled “Adaptive partitioning of a gene locus to the nuclear envelope driven by polymer-polymer phase separation” investigate the mechanisms by which subnuclear localization of genes from nucleoplasm to the nuclear envelope in yeast *S. cerevisiae* by utilizing the inositol depletion-responsive gene, *INO1*. The authors validated earlier findings showing that the active *INO1* locus associates with the nuclear periphery and that deletion of GRS I and GRS II sequences prevent localization of the active *INO1* locus to the nuclear envelope. Next, they measured the state of compaction at the *INO1* locus based on the differences in the mechanical stiffness of the locus in the silent and active states and provided evidence for stiffening of decompacted chromatin at the *INO1* when the locus is active in wild-type cells and in cells in which GRS I or GRS II were individually mutated. Examining the rate and amplitude of *INO1* locus-contraction, it was shown that the locus changes shape with higher frequencies and amplitudes in the repressed state compared to the activated state. Activation of *INO1* lead to increase in the rates of histone H3 exchange. By taking a candidate approach, the authors next utilized AID system to deplete 12 known HAT catalytic subunits. Among these factors, depletion of Gcn5, Rtt109 and Taf1 lead to reduced partitioning of the active *INO1* towards the nuclear envelope suggesting a potential role for these proteins in chromatin remodeling during decompaction of the *INO1* locus. By using bead-spring models of chromatin, the authors examined whether phase transitions might contribute to the partitioning of the *INO1* locus to the nuclear envelope. Their results suggest that similar to experimental conditions, the active *INO1* locus show increased stiffness under stimulated conditions. Based on these findings the authors suggest that the partitioning of gene loci to the nuclear envelope is driven by polymer-polymer phase separation. The findings of the paper are interesting, and the study is timely. However, further analysis and major revision will be necessary to provide more insights on the specific HAT based mechanisms and involvement of phase separation in partitioning of gene loci to the

envelope. I'd like to suggest a major revision with the following changes and additions to strengthen and improve the manuscript:

Major points:

1. Three HAT catalytic subunits including Gcn5, Rtt109 and Taf1 were identified to impact partitioning of the active *INO1*. To determine direct function of these proteins in histone positioning, it is advisable to utilize AID-strains of each protein and perform H3-ChIP-qPCR for specific loci as in Fig. 3.

We thank the reviewer for this advice. Showing more biochemical evidence that these HATs could be involved in the *INO1* locus biophysical changes and localization will better support the results of this study.

The reviewer is suggesting measuring histone exchange at the active *INO1* locus in conditions where the HATs are depleted. While we agree that such data would be compelling, two main challenges make these experiments too complex to achieve. Instead, we performed an alternative that provides evidence for the implication of HATs in *INO1* locus decompaction. We explain the difficulties to perform what the reviewer proposed. First, it would require combining all three induction steps, namely 1, inducing exogenous histones (galactose), 2, inducing the *INO* genes (inositol condition) and, 3, inducing HATs depletions (AID system), all of that while keeping cells arrested in G1 (alpha factor). In our experience, this would have been too difficult to get a reliable result. This is mainly because the inositol response mainly occurs in cells in early mid-log phase. This made it a real challenge to combine *INO1* induction with histone exchange (as in current Fig. 3). Adding another layer (HATs depletion) would be unattainable. Moreover, to create the strains for this assay we would have needed to insert the TIR1 gene and tag each HAT with the degron tag (IAA tag) for the AID system in the histone exchange rate strains. Since these strains and the corresponding GRS mutants already

contain all but one available selection marker that could be used, it would have been also too complex to generate them.

To provide more direct evidence for the implication of the HATs, we instead measured the effect of depletion of each of the three HATs on H3 acetylation in the regions surrounding the *INO1* locus, including the GRS sites, as suggested by the reviewer in point 2 below.

2. In addition to H3-ChIP, changes in H3 acetylation at the related locus should be investigated by CHIP in control and in mutant *Gcn5*, *Rtt109* and *Taf1* strains.

Changes in specific H3 acetylation at the *INO1* and proximal loci were measured as recommended by the reviewer.

The following paragraphs were added to the Results section at line 256 of the revised manuscript:

“We analyzed the effect of the depletion of each of these three HATs on a specific acetylation mark in the regions surrounding the *INO1* locus including the GRS sites. The H3K14 acetylation is known to be catalyzed by *Gcn5*⁷⁻¹⁰, and it has been observed *in vitro* that *Taf1* also acetylates H3 K14¹¹, although this is more controversial¹². *Rtt109* has been also observed to acetylate H3K14 in *in vitro* assays^{13,14}. This mark was quantified by CHIP-qPCR on the *INO1* locus and on proximal genes in the AID strains for each of the three HATs. Results are presented as the ratio between each mark and H3 levels in each region analyzed. The regions screened correspond to the same ones used when measuring the rate of incorporation of newly synthesized histones (Fig. 3).

The results obtained (Fig. 4h), show that there is a significant decrease in the H3K14ac/H3 ratio for the strain where Gcn5 was depleted compared to wild type in all regions analyzed. A significant decrease is also observed for the other two HATs, mainly for the regions close to the GRSII site, however not as significant as for the Gcn5-AID strain. This result indicates that Gcn5 could be the dominant HAT for acetylation in these regions.”

A new panel was added to Fig. 4, panel h, with the corresponding text in the legend of the figure.

Fig. 4, new panel h legend reads:

“H3K14ac enrichment around and within the active *INO1* gene, as determined by CHIP-qPCR and expressed as H3K14ac/H3 ratio, in the vicinity of *DAS1*, *SNA3*, *INO1* and *VPS35* gene loci for wild type and three different HAT-AID strains treated with auxin and under inositol activating conditions. Data represent mean \pm SD (h and i); n = 3 biological samples. Data were analyzed by unpaired

two-sided Student's t-test and p -values are shown for significant differences between wild type and Gcn5-AID mutant."

The following paragraph was added to the Methods section lines 843 of the revised manuscript:

"ChIP-qPCR experiments for H3K14ac and H3 in the AID strains were performed in three biological replicates similarly as described before for the H3-FLAG with modifications for the sample preparation and the IP procedures. In this case, cells were grown in the same conditions as described for the AID experiments before crosslinking. For the IP protocols, 50 μ l of Dynabeads™ Protein G beads (ThermoFisher Scientific) pre-coupled to 2 μ l of rabbit anti-H3K14ac (Millipore) or 2 μ g of rabbit anti-H3 (Abcam) were used per IP sample."

We attempted to measure H3 K56 acetylation that is catalyzed by Rtt109¹⁵⁻¹⁸, but our efforts were thwarted by poor quality of results with the available commercial antibody, including one prepared from a separate lot. We did, however observe reduced H3K14 acetylation when Rtt109 was depleted for some of the regions analyzed upstream the *INO1* gene, suggesting that either Rtt109 can acetylate these sites or that acetylation of H3 K56 exerts an indirect effect on acetylation of H3 K14 by other HATs.

3. Transcription of *INO1* and flanking genes (*SNA3*, *DAS1*, *VP35*) should be quantified in Gcn5, Rtt109 and Taf1 mutant strains. Given that Rtt109 and Taf1 are newly identified molecular regulators of *INO1* partitioning, it is crucial that their function at the spatial partitioning as well as transcription should be determined in detail.

As recommended for the reviewer we evaluated transcription by measuring mRNA levels for the *INO1* gene and proximity genes in the three HAT-AID strains mentioned.

The following paragraph was added in line 272, Results section of the revised manuscript.

“We quantified mRNA levels for these loci after depleting each of the HATs in the AID strains. We observed a decrease in *INO1* mRNA levels for Gcn5-AID and Rtt109-AID depletion strains, and a decrease for *SNA3* mRNA levels, in this case for all three HAT-AID strains, although more significantly for Gcn5-AID and Rtt109-AID strains (Fig. 4i). For the *DAS1* and *VPS35* loci, there was no apparent difference in mRNA levels between all four strains (Fig. 4i). These results indicate that the transcription is affected for *INO1* and the GRS containing upstream locus *SNA3*, when at least two of the three HATs are depleted. Previous results have shown that deleting the GRS sites cause a decrease in *INO1* mRNA levels¹⁹.”

A new panel was added to Fig. 4, panel i, with the corresponding text in the legend of the figure of the revised manuscript.

Fig. 4, new panel 4i legend reads:

“i mRNA levels as quantified by RT-qPCR, relative to *ACT1*, for *DAS1*, *SNA3*, *INO1* and *VPS35* loci in wild type and Gcn5-AID, Taf1-AID, and Rtt109-AID strains in the

presence of auxin and *INO1* activating conditions. Data represent mean \pm SD (h and i); n = 3 biological samples. Data were analyzed by unpaired two-sided Student's t-test and *p*-values are shown for significant differences between wild type and Gcn5-AID mutant.”

The following Section text was added to the Methods at line 850 of the revised manuscript:

“Reverse transcriptase real-time quantitative PCR (RT-qPCR)

Cells were prepared under the same conditions described before for AID and inositol starvation. Total RNA was isolated from yeast cells using TRIzol reagent (Invitrogen) following manufacturer's instructions. Approximately 2 μ g of total RNA was reverse transcribed using All-In-One 5X RT MasterMix (abm). qPCR reactions were performed using FastStart Essential DNA Green Master (Roche Diagnostics) and gene expression levels were determined with a LightCycler 96 (software version 1.1). All assays were performed in biological triplicates.”

4. While modeling to explore the physical properties of chromatin fiber and mechanisms responsible for phase separation is informative, this data is still correlative and requires experimental data to support the final conclusions of the paper.

We agree with the reviewer that the mechanisms responsible for phase separation is informative. As we write in the Discussion section, there is a remarkable consistency of analytical theory of phase separation of polymers with interacting ligands and recent experimental evidence of histone-acetylation-mediated phase separation of nucleosome arrays *in vitro*. As we state at line 387:

“Specifically, acetylation-deacetylation of the histone tails in nucleosomal arrays has been shown to underlie PPPS in which deacetylated histones induce formation of a protein-DNA condensate and acetylation causes its dissolution ^{1.}”

Like all models, however, the purpose of the modeling is to provide intuition regarding the physical principles that could govern the special dynamics of chromatin in a constrained and viscous environment, one that we otherwise have little intuition for. The model reveals that phase separation is sufficient to drive the translocation of the active PPPS to the nuclear envelope. While it is likely that there are active processes that contribute to the dynamics of chromatin, these act in random directions and are implicit in our model. The model brings a critical perspective for expecting the active PPPS in this position. We have already partly addressed this point in the Results section, line 339, of the revised manuscript.

“Bead-spring models of chromatin provide the means to explore the physical properties of the chromatin fiber and mechanisms responsible for polymer, including DNA phase separation. As predicted by Flory-Huggins theory^{2,3}, invoking dynamic chromosome cross-linkers (bound ligand) into a model of the yeast genome recapitulates the segregation and morphology of the nucleolus⁴. The physical basis for partitioning the *INO1* gene locus extends beyond recruitment to the NE. The radius of confinement and mobility of the locus are decreased, indicative of a change in the polymeric state that accompanies its migration to the NE. While we can readily identify transcriptionally active domains relative to DNA sequence, we are just starting to appreciate how gene activity manifests in structural changes at the chromosome level. Active genes can be sequestered into chromosome loops (topologically associated domains) or into membrane-less compartments ^{1,5.}”

Minor points:

1. Figure annotations: Figure annotations (Fig. 2.1-2.5) for all figures need to be corrected.

We thank the reviewer for pointing this out, we have corrected all figure labels in the manuscript.

2. Lines 123-129: This section could be included under discussion.

We considered this was more relevant to mention in this section since we do not further discuss these results.

3. Lane 138: Indicate which data/fig is discussed in this section. If Figure is missing, please include.

In this case we refer to the fact that, as was observed for the *GAL1-10* genes²⁰, once the gene is activated it is found to be distributed towards the NE, but under repressing conditions, it can explore larger regions of the nucleoplasm constrained only by telomere and centromere tethering. These are illustrated by trajectories of active or repressed *INO1* loci:

a representative photomicrographs of yeast nuclei containing the LacO repressor array integrated into the *INO1* locus and expressing GFP-LacI, allowing for visualization of dynamics in the locus with GFP-Nup49-labeled NE and Nop1-mCherry-labeled nucleolus as spatial reference markers. **b** representative trajectories of *INO1* locus under active or repressed conditions.

4. Fig. 3: Including a supplementary figure showing a schematic of genes and marking sites at which H3 occupancy was quantified using ChIP-qPCR would be informative.

A supplementary figure was included as recommended:

Supplementary Fig. 1 legend reads:

“*INO1* and proximal loci positions in chromosome X. Black bars indicate positions where the incorporation of newly synthesized FLAG-H3 into nucleosomes was measured by ChIP-qPCR assays.”

5. Fig. 3: Fig 3c is missing p values.

We removed Fig. 3 panel c since panel d contains the same results but includes the GRSI, I1Δ mutant. *p*-values from Student’s t-tests are shown for the regions with significant differences between the GRS double mutant and the wild type active gene.

6. Lane271-285. The paragraph should be revised, and related Tables and supplementary tables should be specifically indicated. In the current version,

Supplementary Tables start from number 4.

The mentioned paragraph and corresponding figures and tables were properly reorganized and edited starting at line 298, of the Results section of the revised manuscript reading:

“We observed enrichment of peptides derived from proteins associated with several nuclear complexes in extracts from strains expressing versus those not expressing sgRNAs (Fig. 5 b-e, Supplementary Fig. 6) (Table 1; Supplementary Data 1-6). Five of the complexes are associated with chromatin remodeling activity including Ino80, Swr1, ASTRA, Set3, and FACT complexes (Table 1). Among the enriched peptides, those belonging to the ATP-dependent DNA helicases Rvb1 and/or Rvb2 were enriched in all separate sgRNA-BirA*-dCas9 samples. Rvb1 and Rvb2 are components of Ino80²¹, Swr1²², and ASTRA²² complexes and they are also linked to two (Gcn5 and Taf1)²³⁻²⁵ of the three HAT catalytic subunits, whose depletion resulted in reduced partitioning of the active *INO1* towards the NE. Specifically, Gcn5 and Taf1 make protein-protein interactions with Rvb1 (Gcn5) and Rvb2 (Gcn5 and Taf1) and Taf1 also interacts with Ino80, another subunit of the Ino80 complex. Both subunits of the FACT complex Spt16 and Pob3 showed peptides enriched in one of the sgRNA-BirA*-dCas9 samples, and there are reported interactions of these proteins with several HATs including Gcn5^{23,26} and Rtt109²⁷.”

Reviewer #3 (Remarks to the Author):

This work entitled “Adaptive partitioning of a gene locus to the nuclear envelope driven by polymer-polymer phase separation” demonstrated that yeast inositol depletion-responsive gene locus *INO1* partitions to the nuclear envelope was driven by local histone acetylation-induced polymer-polymer phase separation from the nucleoplasmic phase. These results explained the mechanism underlying relocalization of active gene loci from nucleoplasm to the NE and is consistent with recent evidence for chromatin phase separation by acetylation-mediated dissolution of multivalent histone association. However, some concerns are required to be addressed before consideration of publication.

Major concerns:

1. In the line 116-117, the author mentioned that "the *INO1* locus was distributed towards, but not fixed at the NE." How the author can make this conclusion, and what is the evidence to distinguish between distribution and fixation.

Previous studies of an active *GAL* gene locus distributions contrasted their behaviors with those of protein complexes associated with the nuclear envelope such as the spindle pole body (SPB), which displays a constrained distribution that colocalizes with the NE. Like the *GAL* locus, the active *INO1* locus shows a constrained distribution near to but not colocalized with the NE. Also tracking the *INO1* gene shows that the active *INO1* locus is close to the NE, exploring a more confined area than when repressed. These are illustrated by trajectories of active or repressed *INO1* loci:

a representative photomicrographs of yeast nuclei containing the LacO repressor array integrated into the *INO1* locus and expressing GFP-LacI, allowing for visualization of dynamics in the locus with GFP-Nup49-labeled NE and Nop1-mCherry-labeled nucleolus as spatial reference markers. *b* representative trajectories of *INO1* locus under active or repressed conditions.

2. In line 188-189, the author mentioned that "Induction of H3-FLAG allows us to measure the degree of incorporation of newly synthesized H3 into nucleosome complexes by ChIP-qPCR", can the author describe the detection procedure and underlying mechanism.

Nucleosomes exchange histones regularly during the cell cycle, and the exchange dynamics for different regions of chromatin vary according to chromatin state and positions on chromosomes^{17,28}. In this study, we used a strain in which H3 and H4 genes are deleted from the genome and replaced with both constitutively and inducible expressed copies. In a galactose inducible cassette, H3 is fused to a FLAG tag that when expressed can be used to quantify newly synthesized H3 that is incorporated into nucleosomes at specific positions by ChIP-qPCR. This serves as a measure of the dynamics of nucleosome exchange in different chromatin regions under different conditions. In our experiments, the quantification was performed by chromatin immunoprecipitation followed by quantitative PCR (ChIP-qPCR) for wild type and mutant strains. To avoid the contribution of histone incorporation due to DNA

replication, cells are also arrested in G1 phase by the addition of α -factor. This assay was used previously by many labs to study histone exchange, both at specific loci and genome-wide^{17,28-32}

We thank the reviewer for pointing this out, we have better explained this procedure in the Results section, line 183:

“To test whether the changes in *INO1* locus mechanical properties follow from an increase in histone exchange, we used a strain to probe changes in histone H3 nucleosome exchange in *wild type* and GRS1, IIA double mutant strains. In this strain histones H3 and H4 genes are deleted from the genome and replaced with both constitutively and inducible expressed copies (Fig. 3a)¹⁷. In a galactose inducible cassette, H3 is fused to a FLAG-tag that when expressed can be used to quantify newly synthesized H3 that is incorporated into nucleosomes at specific positions by Chromatin Immunoprecipitation followed by real time PCR (ChIP-qPCR) to measure nucleosome exchange in different chromatin regions under different conditions. To avoid the contribution of histone incorporation due to DNA replication, cells are also arrested in G1 by the addition of α -factor. We probed H3-FLAG incorporation in several regions in the *INO1* promoter and ORF, plus sites in upstream (*SNA3* and *DAS1*) and downstream (*VPS35*) genes to determine whether changes in histone exchange are confined to the *INO1* locus (Fig. 3b, Supplementary Fig. 1).”

3. To probe the HATs and other chromatin remodeling complexes proximal to specific GRSs, the author developed a BioID approach based on sgRNA-directed catalytically dead (d)Cas9. Related proximity proteomic approaches have been well demonstrated recently. But does the authors have related data to support if the introduction of (d)Cas9 would cause some stereoscopic hindrance, which might influence the interaction of chromatin remodeling proteins with GRS sites.

Steric constraint and hindrance cannot be ruled out as remarked by the reviewer, and we have inserted a mention about the matter in the text (line 311 of the revised manuscript and below). We removed two sgRNA-BirA*-dCas9 samples out of a total of 4 because they did not reveal significant fold changes for many or most of the candidates that were identified in the samples kept in this revised version of the manuscript, supporting hindrance or constraint to some various degree in relation to either gRNA localization on chromatin, BirA*-dCas9 fusion orientation and linker length. In line, several proteins also vary in enrichment between samples as shown in the new Supplementary Data 4-6 and Supplementary Fig. 6 described in the next point, which also support some degree of steric label constraint or hindrance.

Sentence added in line 311, Results section of the revised manuscript:

“Analysis of protein enrichment in the samples expressing sgRNAs revealed that some proteins were only enriched in one of the samples, or the enrichment of several proteins varied between samples. This may indicate steric label constraint or hindrance due to sgRNA localization on chromatin, BirA*-dCas9 fusion orientation and linker length (Supplementary Fig. 6, Supplementary Data 4-6).”

4. In the LC-MS data (Figure 4i), the author didn't perform the quality control evaluation to estimate the reproducibility of three replicates. And the threshold of significant protein filtration through volcano plot only provided the difference (1.5 fold), but the FDR was not provided.

In the revised version of the manuscript, we have redone the analysis with unique peptides only, and with the quantification condition of not having any missing values in the biological triplicates expressing sgRNAs.

We agree that merging results of all samples on a volcano plot was far from optimal, and we thank the reviewer for pointing this out. We have separated the samples for plotting and replaced volcano plots with micro array (MA) plots, on which average intensities are plotted against fold change while the data points are color coded to reflect fold change p -values obtained from a two-sided unpaired t-test with equal variance from biological triplicates. We also added box plots to show sample variation and visualize significances in fold changes. To present this we have created a separate figure for the MS results, which is now Fig. 5. Highlighted proteins in the plots presented in the new Fig. 5 are subunits of chromatin remodeling complexes that were enriched in the sgRNA containing samples, although we do not ignore other candidates identified. We additionally plotted individual protein intensities for all triplicates and added standard deviations, but these were not added in this revised version of the manuscript since they look identical to the boxplots presented here. We have also updated Table 1. Supplementary Data 1 and 2 are now excel files, avoiding column truncations that rendered the previous tables difficult to read. They represent the raw data for the two samples kept in this revised version of the manuscript and their respective negative controls (S5 vs S3, S7 vs S2), and Supplementary Data 3 and 4 show all protein measurements that passed the quantification conditions as described in the text added at line 928 as mentioned below. Supplementary Data 3 and 4 also contain standard deviations, variation coefficient as well as fold changes and associated p -values. Moreover, we have cross-checked these datasets to a list of proteins related to chromatin and transcription obtained from UniProt (Supplementary Data 6), and we now present the output of these cross-checked datasets in Supplementary Data 5. Intensity measurements from proteins in this last table are plotted in the new Supplementary Fig. 6 as box plots. Table 2 was removed since Supplementary Data 5 is more informative.

Paragraph added in line 928, Methods section of the revised manuscript:

“Unique peptides were used for quantification. Only proteins for which values from all biological triplicates were measured in the sgRNA-BirA*-dCas9 were kept for quantification, and no values were imputed during the analysis. The resulting protein datasets were crosschecked with a list of chromatin and transcription related proteins obtained from Uniprot (Supplementary Data 5). The protein list and associated ontologies was built from searches using the keyword chromatin (461 reviewed entries), and transcription (1211 reviewed entries) in *S. cerevisiae*. After merging both lists and removing duplicates, 1320 entries remained (Supplementary Data 6). A two-sided unpaired T-test with equal variance was used to help in the evaluation of significant protein enrichment from sgRNA-BirA*-dCas9 triplicates. Analyses of evidence for association of chromatin remodeling subunits with nuclear protein complexes was performed for all significant candidates with definitions of subunit compositions of protein complexes taken from the Saccharomyces Genome Database (SGD) ³³ and UniProt ³⁴, and protein-protein interactions were identified in The Biological General Repository for Interaction Datasets (BioGrid) ³⁵. For representation clarity, proteins having a log2 fold change above 2.5 were limited at the 2.5 value on MA plots (Fig. 5). We removed two sgRNA-BirA*-dCas9 samples out of a total of 4 because they did not reveal significant fold changes for many or most of the candidates that were identified in the samples kept for further analysis. Both MA and box plots were made in RStudio (v2022.07.2) using the ggplot2 package (v3.3.6).”

Changes added in line 298, Results section of the revised manuscript:

“We observed enrichment of peptides derived from proteins associated with several nuclear complexes in extracts from strains expressing versus those not expressing sgRNAs (Fig. 5b-e, Supplementary Fig. 6) (Table 1; Supplementary

Data 1-6). Five of the complexes are associated with chromatin remodeling activity including Ino80, Swr1, ASTRA, Set3, and FACT complexes (Table 1). Among the enriched peptides, those belonging to the ATP-dependent DNA helicases Rvb1 and/or Rvb2 were enriched in all separate sgRNA-BirA*-dCas9 samples. Rvb1 and Rvb2 are components of Ino80²¹, Swr1²², and ASTRA²² complexes and they are also linked to two (Gcn5 and Taf1)²³⁻²⁵ of the three HAT catalytic subunits, whose depletion resulted in reduced partitioning of the active *INO1* towards the NE. Specifically, Gcn5 and Taf1 make protein-protein interactions with Rvb1 (Gcn5) and Rbv2 (Gcn5 and Taf1) and Taf1 also interacts with Ino80, another subunit of the Ino80 complex. Both subunits of the FACT complex Spt16 and Pob3 showed peptides enriched in one of the sgRNA-BirA*-dCas9 samples, and there are reported interactions of these proteins with several HATs including Gcn5^{23,26} and Rtt109²⁷."

Using a fixed protein fold enrichment threshold may mask significant protein candidates. For example, a small change for an abundant protein may result in significant change in intensity as we are showing for Cpr1 on the MA and box plots in this revised version of the manuscript (see Fig. 5b-e). Yeast endogenous biotinylation should also be kept in mind along with steric hindrance / constraints as discussed above, which makes the use of a specific fold enrichment threshold difficult.

An FDR threshold of 0.01 was used at the peptide spectrum match, as well as peptide and protein levels as indicated in the material and methods, but we now also specify this threshold in the figure legend.

a**b****c****d****e**
Fold Change p-value

- $p < 0.01$
- $0.01 < p < 0.05$
- $0.05 < p < 0.10$
- $p > 0.10$

BirA*-dCas9-gRNA

BirA*-dCas9

New Fig. 5 legend:

“Fig. 5 Chromatin remodeling subunits bind GRS site near *INO1* locus.

a BirA*-dCas9 system generated to target the vicinity of GRS I site by using different sgRNAs for site-specific binding of BirA*-dCas9 and biotinylation of proximal proteins. Proteins were then affinity purified with streptavidin, digested, and analyzed by mass spectrometry. **b, d**, Micro array (MA) plots representing average protein intensities from biological triplicates from sgRNA-BirA*-dCas9 samples S5 (**b**) and S7 (**c**) plotted against protein quantity fold changes from these sgRNA-BirA*-dCas9 samples over those from their respective negative control triplicates (BirA*-dCas9). Protein fold change *p*-values calculated from a two-sided unpaired T-tests with equal variance are color coded, and the grey dashed lines represent 1.5-fold quantity increase. *p*-values are indicated next to labels. **c, e**, Box plots of measured protein intensities of candidates from sgRNA-BirA*-dCas9 samples S5 (**d**) and S7 (**e**) triplicates along with their respective negative control triplicates. Median is indicated as middle line, average as a white dot, 25th and 75th percentile as boxes and whiskers represent 5th and 95th percentile. False discovery rates thresholds for mass spectrometry identification of peptide spectrum match, peptides, and proteins for were set to 0.1. See also Supplementary Data 1-6 and Supplementary Fig. 5 and 6.”

New Supplementary Data 1-6:

Supplementary Data 1: Raw Spectronaut output of the mass spectrometry analysis between biological triplicates of BirA*-l2-dCas9-u2 (samples S5) and corresponding negative controls BirA*-l2-dCas9 (samples S3).

Supplementary Data 2: Raw Spectronaut output of the mass spectrometry analysis between biological triplicates of BirA*-l1-dCas9-d3 (samples S7) and corresponding negative controls BirA*-l1-dCas9 (samples S2).

Supplementary Data 3: Protein candidates meeting quantification requirements as described in the mass spectrometry method section for samples 5 and 3 biological triplicates, along with their intensities, average intensities, standard deviations, variation coefficients, fold changes, and associated *p*-values.

Supplementary Data 4: Protein candidates meeting quantification requirements as described in the mass spectrometry method section for samples 7 and 2 biological triplicates, along with their intensities, average intensities, standard deviations, variation coefficients, fold changes, and associated *p*-values.

Supplementary Data 5: Protein candidates list from both mass spectrometry datasets matching the Uniprot chromatin and transcription related proteins list, along with their ontologies.

Supplementary Data 6: Uniprot search output of chromatin and transcription related proteins in *S. cerevisiae*, along with their ontologies.

Higher abundance proteins

Samples and Proteins

Chromatin / chromatin modifiers

Samples and Proteins

Transcriptional regulation

Samples and Proteins

mRNA regulation / transport

Samples and Proteins

New Supplementary Fig. 6 legend:

“Box plots of proteins intensities measured by MS for all proteins related to chromatin or transcription meeting quantification criteria. Purified protein measurements from biological triplicates expressing either gRNAs-BirA*-dCas9 (S5 and S7; +) along with their respective BirA*-dCas9 negative controls (S5 and S7; -) are shown side by side. Median is indicated as middle line, average as a black dot, 25th and 75th percentile as boxes and whiskers represent 5th and 95th percentile. For plotting, proteins were grouped by function, and proteins with significant higher abundance compared to their corresponding group were plotted separately. See also Supplementary Data 4-6.”

5. Along this line, the proteomics data presentation should be significantly improved. For example, the authors should perform a general bioinformatic analysis for the significantly changed proteins and demonstrate that significant number of proteins have function relevance, rather the random highlighted proteins. In addition, Table 1 and 2 only highlighted the protein names and function annotation. The authors should provide related quantitative proteomic data to support their relevance. Last but not the least, the authors should provide extended supporting figure presentation to present the basic proteomic data analysis, such as identified proteins/peptides in each replicate, the reproducibility evaluation, etc.

Data representation and analyses have been addressed in point 4 above. We noticed that the raw export tables were truncated by groups of columns, which made the tables unreadable. These tables have been fixed and replaced (Supplementary Data 1 and 2) along with protein lists and individual quantities, average intensities, standard deviations, variation coefficients, fold change and fold change p-values (Supplementary Data 3 and 4) for proteins meeting quantification requirements as mentioned before.

Also as mentioned in point 4, highlighted proteins in the plots presented in the new Fig. 5 are subunits of chromatin remodeling complexes that were enriched in the sgRNA containing samples, although we do not ignore the importance of other candidates identified. Moreover, as mentioned, we have crosschecked this list of candidates that met quantification requirements to a list of proteins related to chromatin and transcription obtained from UniProt and checked for their binding to different nuclear complexes (Supplementary Data 5 and 6). The new Supplementary Fig. 6, described in point 4, shows box plots of all these chromatin and transcription related candidates identified in both samples kept in this revised version of the manuscript.

6. Although the author identified some chromatin remodeling proteins through proximity labeling, these results were not confirmed through biochemical results, and related publications which can support these interactions were not cited, which might make the conclusion not that convincing to readers.

The identification of chromatin remodeling proteins with the proximity labeling assay supports our results for histone exchange and the disruption of the localization when depleting specific HATs. We agree with the reviewer that more biochemical data related to this would support better the results of this study. Since these chromatin remodelling subunits identified can interact with several complexes including HATs shown to disrupt the locus localization when depleted, we therefore analyzed specific acetylation of H3 at and surrounding the *INO1* locus (Fig. 4h). We also quantified transcription of *INO1* and flanking genes (*SNA3*, *DAS1*, *VP35*) in Gcn5, Rtt109 and Taf1 AID mutant strains (Fig. 4i).

H3K14ac quantification at *INO1* and proximity genes in HAT-AID strains:

The following paragraphs were added to the Results section line 256 of the revised manuscript.

“We analyzed the effect of the depletion of each of these three HATs on a specific acetylation mark in the regions surrounding the *INO1* locus including the GRS sites. The H3K14 acetylation is known to be catalyzed by Gcn5⁷⁻¹⁰, and it has been observed *in vitro* that Taf1 also acetylates H3 K14¹¹, although this is more controversial¹². Rtt109 has been also observed to acetylate H3K14 in *in vitro* assays^{13,14}. This mark was quantified by CHIP-qPCR on the *INO1* locus and on proximal genes in the AID strains for each of the three HATs. Results are presented as the ratio between each mark and H3 levels in each region analyzed. The regions screened correspond to the same ones used when measuring the rate of incorporation of newly synthesized histones (Fig. 3).

The results obtained (Fig. 4h), show that there is a significant decrease in the H3K14ac/H3 ratio for the strain where Gcn5 was depleted compared to wild type in all regions analyzed. A significant decrease is also observed for the other two HATs, mainly for the regions close to the GRSII site, however not as significant as for the Gcn5-AID strain. This result indicates that Gcn5 could be the dominant HAT for acetylation in these regions.”

A new panel was added to Fig. 4, panel h, with the corresponding text in the legend of the figure, at the revised manuscript.

Fig. 4, new panel 4h legend reads:

“H3K14ac enrichment around and within the active *INO1* gene, as determined by CHIP-qPCR and expressed as H3K14ac/H3 ratio, in the vicinity of *DAS1*, *SNA3*, *INO1* and *VPS35* gene loci for wild type and three different HAT-AID strains treated with auxin and under inositol activating conditions. Data represent mean \pm SD (h and i); $n = 3$ biological samples. Data were analyzed by unpaired two-sided Student’s t-test and p -values are shown for significant differences between wild type and Gcn5-AID mutant.”

The following paragraph was added to the Methods section lines 843 of the revised manuscript:

“CHIP-qPCR experiments for H3K14ac and H3 in the AID strains were performed in three biological replicates similarly as described before for the H3-FLAG with modifications for the sample preparation and the IP procedures. In this case, cells were grown in the same conditions as described for the AID experiments before crosslinking. For the IP protocols, 50 μ l of Dynabeads™ Protein G beads

(ThermoFisher Scientific) pre-coupled to 2 μ l of rabbit anti-H3K14ac (Millipore) or 2 μ g of rabbit anti-H3 (Abcam) were used per IP sample.”

We attempted to measure H3 K56 acetylation that is catalyzed by Rtt109¹⁵⁻¹⁸, but our efforts were thwarted by poor quality of results with the available commercial antibody, including one prepared from a separate lot. We did, however observed reduced H3K14 acetylation when Rtt109 was depleted for some of the regions analyzed upstream the *INO1* gene, suggesting that either Rtt109 can acetylate these sites or that acetylation of H3 K56 exerts an indirect effect on acetylation of H3 K14 by other HATs.

mRNA quantification at *INO1* and proximity genes in HAT-AID strains:

The following paragraph was added in line 272, Results section of the revised manuscript.

“We quantified mRNA levels for these loci after depleting each of the HATs in the AID strains. We observed a decrease in *INO1* mRNA levels for Gcn5-AID and Rtt109-AID depletion strains, and a decrease for *SNA3* mRNA levels, in this case for all three HAT-AID strains, although more significantly for Gcn5-AID and Rtt109-AID strains (Fig 4i). For the *DAS1* and *VPS35* loci, there was no apparent difference in mRNA levels between all four strains (Fig. 4i). These results indicate that the transcription is affected for *INO1* and the GRS containing upstream locus *SNA3*, when at least two of the three HATs are depleted. Previous results have shown that deleting the GRS sites cause a decrease in *INO1* mRNA¹⁹.”

A new panel was added to Fig. 4, panel i, with the corresponding text in the legend of the figure, at the revised manuscript.

Fig. 4, new panel 4i:

“i mRNA levels as quantified by RT-qPCR, relative to *ACT1*, for *DAS1*, *SNA3*, *INO1* and *VPS35* loci in wild type and Gcn5-AID, Taf1-AID, and Rtt109-AID strains in the presence of auxin and *INO1* activating conditions. Data represent mean \pm SD (h and i); n = 3 biological samples. Data were analyzed by unpaired two-sided Student’s t-test and *p*-values are shown for significant differences between wild type and Gcn5-AID mutant.”

The following Section text was added to the Methods at line 850 of the revised manuscript:

“Reverse transcriptase real-time quantitative PCR (RT-qPCR)

Cells were prepared under the same conditions as the ones described before for AID system and inositol starvation. Total RNA was isolated from yeast cells using TRIzol reagent (Invitrogen) following manufacturer's instructions. Approximately

2 µg of total RNA was reverse transcribed using All-In-One 5X RT MasterMix (abm). qPCR reactions were performed using FastStart Essential DNA Green Master (Roche Diagnostics) and gene expression levels were determined with a LightCycler 96 (software version 1.1). All assays were performed in biological triplicates.”

Finally, references that support enriched proteins identified in the MS experiment binding to the HATs observed to disrupt *INO1* localization when depleted are now shown in the text.

Minor concern:

The figure labels in the manuscript are not corresponding to the figure part.

We thank the reviewer for pointing this out, we have corrected all figure labels in the manuscript.

- 1 Gibson, B. A. *et al.* Organization of Chromatin by Intrinsic and Regulated Phase Separation. *Cell* **179**, 470-484.e421, doi:10.1016/j.cell.2019.08.037 (2019).
- 2 Flory, P. J. & Gee, G. Statistical thermodynamics of semi-flexible chain molecules. *Proceedings of the Royal Society of London. Series A. Mathematical and Physical Sciences* **234**, 60-73, doi:10.1098/rspa.1956.0015 (1956).
- 3 Huggins, M. L. THERMODYNAMIC PROPERTIES OF SOLUTIONS OF LONG-CHAIN COMPOUNDS. *Annals of the New York Academy of Sciences* **43**, 1-32, doi:<https://doi.org/10.1111/j.1749-6632.1942.tb47940.x> (1942).
- 4 Hult, C. *et al.* Enrichment of dynamic chromosomal crosslinks drive phase separation of the nucleolus. *Nucleic Acids Res* **45**, 11159-11173, doi:10.1093/nar/gkx741 (2017).
- 5 Sanulli, S. *et al.* HP1 reshapes nucleosome core to promote heterochromatin phase separation. *Nature*, doi:10.1038/s41586-019-1669-2 (2019).
- 6 Jun, S. & Mulder, B. Entropy-driven spatial organization of highly confined polymers: lessons for the bacterial chromosome. *Proceedings of the National Academy of Sciences of the United States of America* **103**, 12388-12393, doi:10.1073/pnas.0605305103 (2006).
- 7 Duan, M. R. & Smerdon, M. J. Histone H3 lysine 14 (H3K14) acetylation facilitates DNA repair in a positioned nucleosome by stabilizing the binding of the chromatin Remodeler RSC (Remodels Structure of Chromatin). *The Journal of biological chemistry* **289**, 8353-8363, doi:10.1074/jbc.M113.540732 (2014).
- 8 Smerdon, M. J. DNA repair and the role of chromatin structure. *Current opinion in cell biology* **3**, 422-428, doi:10.1016/0955-0674(91)90069-b (1991).

- 9 Kuo, M. H., vom Baur, E., Struhl, K. & Allis, C. D. Gcn4 activator targets Gcn5 histone acetyltransferase to specific promoters independently of transcription. *Molecular cell* **6**, 1309-1320, doi:10.1016/s1097-2765(00)00129-5 (2000).
- 10 Kuo, M. H. *et al.* Transcription-linked acetylation by Gcn5p of histones H3 and H4 at specific lysines. *Nature* **383**, 269-272, doi:10.1038/383269a0 (1996).
- 11 Mizzen, C. A. *et al.* The TAF(II)250 subunit of TFIID has histone acetyltransferase activity. *Cell* **87**, 1261-1270, doi:10.1016/s0092-8674(00)81821-8 (1996).
- 12 Durant, M. & Pugh, B. F. Genome-wide relationships between TAF1 and histone acetyltransferases in *Saccharomyces cerevisiae*. *Molecular and cellular biology* **26**, 2791-2802, doi:10.1128/mcb.26.7.2791-2802.2006 (2006).
- 13 Berndsen, C. E. *et al.* Molecular functions of the histone acetyltransferase chaperone complex Rtt109-Vps75. *Nature structural & molecular biology* **15**, 948-956, doi:10.1038/nsmb.1459 (2008).
- 14 Abshiru, N. *et al.* Chaperone-mediated acetylation of histones by Rtt109 identified by quantitative proteomics. *Journal of proteomics* **81**, 80-90, doi:10.1016/j.jprot.2012.09.026 (2013).
- 15 Driscoll, R., Hudson, A. & Jackson, S. P. Yeast Rtt109 Promotes Genome Stability by Acetylating Histone H3 on Lysine 56. *Science (New York, N.Y.)* **315**, 649-652, doi:10.1126/science.1135862 (2007).
- 16 Han, J. *et al.* Rtt109 acetylates histone H3 lysine 56 and functions in DNA replication. *Science (New York, N.Y.)* **315**, 653-655, doi:10.1126/science.1133234 (2007).
- 17 Rufiange, A., Jacques, P. E., Bhat, W., Robert, F. & Nourani, A. Genome-wide replication-independent histone H3 exchange occurs predominantly at promoters and implicates H3 K56 acetylation and Asf1. *Molecular cell* **27**, 393-405, doi:10.1016/j.molcel.2007.07.011 (2007).
- 18 Topal, S., Vasseur, P., Radman-Livaja, M. & Peterson, C. L. Distinct transcriptional roles for Histone H3-K56 acetylation during the cell cycle in Yeast. *Nature communications* **10**, 4372, doi:10.1038/s41467-019-12400-5 (2019).
- 19 Ahmed, S. *et al.* DNA zip codes control an ancient mechanism for gene targeting to the nuclear periphery. *Nature cell biology* **12**, 111-118, doi:10.1038/ncb2011 (2010).
- 20 Cabal, G. G. *et al.* SAGA interacting factors confine sub-diffusion of transcribed genes to the nuclear envelope. *Nature* **441**, 770-773, doi:10.1038/nature04752 (2006).
- 21 Shen, X., Mizuguchi, G., Hamiche, A. & Wu, C. A chromatin remodelling complex involved in transcription and DNA processing. *Nature* **406**, 541-544, doi:10.1038/35020123 (2000).
- 22 Nguyen, V. Q. *et al.* Molecular architecture of the ATP-dependent chromatin-remodeling complex SWR1. *Cell* **154**, 1220-1231, doi:10.1016/j.cell.2013.08.018 (2013).
- 23 Lee, K. K. *et al.* Combinatorial depletion analysis to assemble the network architecture of the SAGA and ADA chromatin remodeling complexes. *Molecular systems biology* **7**, 503, doi:10.1038/msb.2011.40 (2011).
- 24 Sanders, S. L., Jennings, J., Canutescu, A., Link, A. J. & Weil, P. A. Proteomics of the eukaryotic transcription machinery: identification of proteins associated with components of yeast TFIID by multidimensional mass spectrometry. *Molecular and cellular biology* **22**, 4723-4738, doi:10.1128/mcb.22.13.4723-4738.2002 (2002).
- 25 Lakshminarasimhan, M. *et al.* Proteomic and Genomic Analyses of the Rvb1 and Rvb2 Interaction Network upon Deletion of R2TP Complex Components. *Molecular & cellular proteomics : MCP* **15**, 960-974, doi:10.1074/mcp.M115.053165 (2016).

- 26 Leng, H. *et al.* FACT interacts with Set3 HDAC and fine-tunes GAL1 transcription in response to environmental stimulation. *Nucleic acids research* **49**, 5502-5519, doi:10.1093/nar/gkab312 (2021).
- 27 Yang, J. *et al.* The Histone Chaperone FACT Contributes to DNA Replication-Coupled Nucleosome Assembly. *Cell reports* **14**, 1128-1141, doi:10.1016/j.celrep.2015.12.096 (2016).
- 28 Dion, M. F. *et al.* Dynamics of replication-independent histone turnover in budding yeast. *Science (New York, N.Y.)* **315**, 1405-1408, doi:10.1126/science.1134053 (2007).
- 29 Jamai, A., Imoberdorf, R. M. & Strubin, M. Continuous histone H2B and transcription-dependent histone H3 exchange in yeast cells outside of replication. *Molecular cell* **25**, 345-355, doi:10.1016/j.molcel.2007.01.019 (2007).
- 30 Jamai, A., Puglisi, A. & Strubin, M. Histone chaperone spt16 promotes redeposition of the original h3-h4 histones evicted by elongating RNA polymerase. *Molecular cell* **35**, 377-383, doi:10.1016/j.molcel.2009.07.001 (2009).
- 31 Radman-Livaja, M. *et al.* A key role for Chd1 in histone H3 dynamics at the 3' ends of long genes in yeast. *PLoS Genet* **8**, e1002811, doi:10.1371/journal.pgen.1002811 (2012).
- 32 Kassem, S. *et al.* Histone exchange is associated with activator function at transcribed promoters and with repression at histone loci. *Sci Adv* **6**, doi:10.1126/sciadv.abb0333 (2020).
- 33 Cherry, J. M. *et al.* Saccharomyces Genome Database: the genomics resource of budding yeast. *Nucleic Acids Res* **40**, D700-705, doi:10.1093/nar/gkr1029 (2012).
- 34 The UniProt, C. UniProt: the universal protein knowledgebase in 2021. *Nucleic Acids Research* **49**, D480-D489, doi:10.1093/nar/gkaa1100 (2021).
- 35 Oughtred, R. *et al.* The BioGRID database: A comprehensive biomedical resource of curated protein, genetic, and chemical interactions. *Protein Sci* **30**, 187-200, doi:10.1002/pro.3978 (2021).